# Unified Graph Augmentations for Generalized Contrastive Learning on Graphs

**Jiaming Zhuo[1], Yintong Lu[1], Hui Ning[1], Kun Fu[1], Bingxin Niu[1], Dongxiao He[2],**
**Chuan Wang[3], Yuanfang Guo[4], Zhen Wang[5], Xiaochun Cao[6], Liang Yang[1]***
[1]Hebei Province Key Laboratory of Big Data Calculation,
School of Artificial Intelligence, Hebei University of Technology, Tianjin, China
[2]College of Intelligence and Computing, Tianjin University, Tianjin, China
[3]School of Computer Science and Technology, Beijing JiaoTong University, Beijing, China
[4]School of Computer Science and Engineering, Beihang University, Beijing, China
[5]School of Artificial Intelligence, OPtics and ElectroNics (iOPEN),
School of Cybersecurity, Northwestern Polytechnical University, Xi'an, China
[6]School of Cyber Science and Technology,
Shenzhen Campus of Sun Yat-sen University, Shenzhen, China
jiaming.zhuo@outlook.com, 202332803037@stu.hebut.edu.cn,
ninghui048@163.com, fukun@hebut.edu.cn, niubingxin666@163.com,
hedongxiao@tju.edu.cn, wangchuan@iie.ac.cn, andyguo@buaa.edu.cn,
w-zhen@nwpu.edu.cn, caoxiaochun@mail.sysu.edu.cn, yangliang@vip.qq.com

## Abstract

In real-world scenarios, networks (graphs) and their tasks possess unique characteristics, requiring the development of a versatile graph augmentation (GA) to meet the varied demands of network analysis. Unfortunately, most Graph Contrastive Learning (GCL) frameworks are hampered by the specificity, complexity, and incompleteness of their GA techniques. Firstly, GAs designed for specific scenarios may compromise the universality of models if mishandled. Secondly, the process of identifying and generating optimal augmentations generally involves substantial computational overhead. Thirdly, the effectiveness of the GCL, even the learnable ones, is constrained by the finite selection of GAs available. To overcome the above limitations, this paper introduces a novel unified GA module dubbed UGA after reinterpreting the mechanism of GAs in GCLs from a message-passing perspective. Theoretically, this module is capable of unifying any explicit GAs, including node, edge, attribute, and subgraph augmentations. Based on the proposed UGA, a novel generalized GCL framework dubbed Graph cOntrastive UnifieD Augmentations (GOUDA) is proposed. It seamlessly integrates widely adopted contrastive losses and an introduced independence loss to fulfill the common requirements of consistency and diversity of augmentation across diverse scenarios. Evaluations across various datasets and tasks demonstrate the generality and efficiency of the proposed GOUDA over existing state-of-the-art GCLs.

## 1 Introduction

Owing to their effectiveness and efficiency, Graph Neural Networks (GNNs) have become a standard toolkit for processing various graph tasks such as node classification and graph classification [17, 34, 41, 40]. They typically follow a message-passing paradigm [10], where the representation of each node is updated by aggregating the representations of its adjacent nodes and subsequently combining

---

*corresponding author

38th Conference on Neural Information Processing Systems (NeurIPS 2024).

the aggregated representations with itself. In general, to produce discriminative representations, GNNs need to resort to the task-relevant labels (*i.e.*, supervised information) to guide the network training, which limits their applicability in the label scarcity scenarios [24, 37, 43]. To overcome this limitation, Graph Contrastive Learning (GCL), a typical graph self-supervised learning architecture, has been developed to provide training guidance by capturing the self-supervised information contained in the graph [53, 32, 50, 2, 21, 39, 20].

Inspired by the design philosophy of contrastive learning in Computer Vision (CV) [4, 12], GCLs adopt the same architecture, which consists of three components: augmentation, encoder, and contrastive loss [53, 32]. Thus, GCLs inherit the merit of enabling learning representations invariant to augmentation, which is achieved by maximizing the agreement between embeddings from different perturbations of the same graph [51, 52]. To further improve the representation capacity of GCLs, great endeavors have been made to design augmentations for the original graph, *i.e.*, Graph Augmentations (GAs), which target nodes, edges, attributes, and subgraphs. Based on how information is processed, GAs can be divided into two categories: *heuristic* [53, 35, 32, 14] and *learnable* methods [47, 31, 21, 56]. The heuristic GAs modify graphs through the combination of fixed, random rules, such as attribute masking [53], edge removing [32, 2], and graph diffusion [14]. They tend to neglect the subsequent steps, namely encoding and contrastive optimization, hence leading to suboptimal performances. In contrast, learnable GAs leverage prior knowledge and feedback during training to refine augmentations, already surpassing base augmentations on many tasks. Notable contributions include GAs based on spectral methods [21], and adversarial training [31].

Given their inherent and distinct characteristics, various networks and tasks require the meticulous selection of optimal GAs to improve model performance pivotally. However, most GCLs face several limitations regarding the selection: (1) *Specificity*. GCLs are typically tailored with specific GAs to meet the needs of particular scenarios, resulting in a lack of generality across diverse scenarios. For instance, node dropping (specifically, removing nodes and their associated edges), widely applied in graph-level tasks [47, 48], could significantly compromise the integrity of graphs [36], rendering it less suitable for node-level tasks. (2) *High complexity*. Either way, identifying and generating the scene-specific GAs impose a considerable computational burden on the models. For example, the set sampling method necessitates a validation of all combinations [45, 38]. Furthermore, the adversarial attack method [47, 31] entails recalculating contrastive losses, which takes a quadratic complexity of $O(n^2)$. Besides, the spectral method requires Laplacian matrix decomposition [5], which has a cubic complexity of $O(n^3)$. (3) *Incompleteness*. Despite the promise of existing learnable GAs in optimizing for specific scenarios, their efficacy is limited by the finite range of GAs at their disposal.

This paper seeks to break these limitations by proposing a unified GA module for GCLs. Toward this end, the mechanisms of existing GAs in GCLs are systemically investigated and reinterpreted from a message-passing perspective [10]. The conclusion is that, from the message-passing perspective, GAs uniformly induce attribute modifications within the neighborhoods of nodes, even though they appear diverse from the spatial perspective, as depicted in Fig. 1. Therefore, the essence of GCLs is to learn node representations invariant to such local augmentation. Drawing from this insight, a novel Unified GA (UGA) module with a simple yet effective design is presented. It strategically interpolates an appropriate amount of Augmentation-Centric (AC) vectors in a graph-structured manner [55, 8], where AC vectors are treated as another type of node, as illustrated in Fig. 2. In theory, UGA is able to simulate the impact of the above four explicit GAs on target nodes by aggregating features from the AC vectors that capture the attribute variations within the neighborhood of these nodes.

Building upon the proposed UGA module, a generalized GCL framework dubbed Graph cOntrastive UnifieD Augmentations (GOUDA) is presented to overcome the above challenges in existing GCLs. This framework adopts a typical dual-channel architecture [4, 49, 53], corresponding to two distinct augmented graphs (views) with their respective AC matrices, as shown in Fig. 2. To realize general utility, GOUDA proposes to capture the consistency and diversity across augmentations (defined in Section 3.3), which are essential and shared goals for GCLs to be applicable across diverse scenarios. To be specific, the objective function of GOUDA is twofold: (1) maximizing Mutual Information (MI) between representations from these distinct views. (2) maximizing the distributional difference between the AC matrices. The former is a fundamental principle behind classic contrastive losses and inherently ensures consistency, while the latter is a constraint to modulate diversity. In practice, GOUDA is instantiated by leveraging widely employed contrastive losses alongside a Hilbert-Schmidt Independence Criterion (HSIC)-based distributional independence loss. This design makes GOUDA more effective and efficient than GCLs with learnable GAs.

The main contributions of this work are summarized as follows:

- We investigate the mechanism of GAs in GCLs through the lens of message-passing.
- We propose a lightweight GA module named UGA to simulate the impacts of GAs on nodes.
- We introduce GOUDA, an efficient and generalized GCL framework, which captures both consistency and diversity across augmentations.
- Extensive experiments and in-depth analysis demonstrate that GOUDA outperforms state-of-the-art GCLs across various public benchmark datasets and tasks.

## 2 Preliminaries

This section briefly introduces the notations utilized throughout the paper. Subsequently, it outlines the essential components of the Graph Contrastive Learning (GCL) framework.

### 2.1 Notations

Matrices (*e.g.*, $\mathbf{Q}$) are in bold capital letters, vectors (*e.g.*, $\mathbf{q}_{i,:}$, which denotes the $i$-th row of $\mathbf{Q}$) are in bold lowercase letters, scalars (*e.g.*, $q_{i,j}$, which represents the entry of $\mathbf{Q}$ at the $i$-th row and the $j$-th column) are in lowercase letters, and sets (*e.g.*, $\mathcal{N}$) are in calligraphic letters.

For a general-purpose description, this paper considers an undirected attribute graph $\mathcal{G}(\mathcal{V}, \mathcal{E})$, where $\mathcal{V}$ stands for the node-set containing $n$ node instances $\{(\mathbf{x}_v, \mathbf{y}_v)\}_{v \in \mathcal{V}}$. And $\mathbf{X} \in \mathbb{R}^{n \times f}$ and $\mathbf{Y} \in \mathbb{R}^{n \times c}$ denote the attribute matrix and label matrix of node $v$, respectively, where $f$ and $c$ is the numbers of attirbutes and labels, respectively. Also, $\mathcal{E} = \{e_i\}_{i=0}^{m-1}$ terms the edge set containing $m$ edges. In general, the adjacency matrix $\mathbf{A} \in \mathbb{R}^{n \times n}$ is employed to describe the graph topology, such that the matrix form of the graph can be expressed as $\mathcal{G}(\mathbf{A}, \mathbf{X})$. Moreover, $\mathbf{H} \in \mathbb{R}^{n \times d}$ terms the graph representation, where $d$ terms the dimension of the representation.

### 2.2 Graph Contrastive Learning

**Graph Augmentations.** Drawing on the successful experience of image augmentation in Computer Vision (CV) [4, 15], Graph Augmentation (GA) [51] is introduced in graph learning to address the challenge of data scarcity. In the typical GCL frameworks, the input graph $\mathcal{G}(\mathbf{A}, \mathbf{X})$ is processed through two separate perturbations (GA procedures), formulated as $\mathrm{t}_i(\mathcal{G}) : \mathcal{G}(\mathbf{A}, \mathbf{X}) \to \mathcal{G}_i(\mathbf{A}^{(i)}, \mathbf{X}^{(i)})$, to generate its two views (augmented graphs), denoted as $\mathcal{G}_1(\mathbf{A}^{(1)}, \mathbf{X}^{(1)})$ and $\mathcal{G}_2(\mathbf{A}^{(2)}, \mathbf{X}^{(2)})$. Based on the perturbed information, GAs can be broadly classified into four main categories: *node augmentation* [47, 48], *edge augmentation* [53, 54, 32, 31], *attribute augmentation* [19, 53, 44], and *subgraph augmentation* [47, 13]. An overview of GAs can be found in Section B.

**Graph Encoders.** For efficient processing and analysis, the graph encoders are leveraged to transform raw topology and attribute information of the input graph into low-dimensional vector representations. Most graph encoders in GCLs follow a message-passing paradigm [10], which typically involves two primary processes: *aggregation* and *combination*. During these steps, each node iteratively updates its representations by aggregating and combining the node features from its neighborhoods, that is

$$\mathring{\mathbf{h}}_v^l \triangleq \mathrm{Aggregation}^l\left(\{\mathbf{h}_u^{l-1} | u \in \mathcal{N}_v\}\right), \quad \mathbf{h}_v^l \triangleq \mathrm{Combination}^l\left(\mathbf{h}_v^{l-1}, \mathring{\mathbf{h}}_v^l\right), \quad (1)$$

where $\mathbf{h}_v^l$ terms the representations of node $v$ in the $l$-th layer and $\mathcal{N}_v$ denotes the set of neighboring nodes of node $v$. In prevalent GCLs like GRACE [53], a two-layer GCN [17] is adopted, where the $\mathrm{Aggregation}(\cdot)$ and $\mathrm{Combination}(\cdot)$ functions are implemented via average function. Thus, there is

$$\mathbf{H} = \mathrm{GCN}^2\left(\mathcal{G}\left(\mathbf{A}, \mathbf{X}\right)\right) = \sigma\left(\hat{\mathbf{D}}^{-\frac{1}{2}}\hat{\mathbf{A}}\hat{\mathbf{D}}^{-\frac{1}{2}}\sigma\left(\hat{\mathbf{D}}^{-\frac{1}{2}}\hat{\mathbf{A}}\hat{\mathbf{D}}^{-\frac{1}{2}}\mathbf{X}\mathbf{W}^0\right)\mathbf{W}^1\right), \quad (2)$$

where $\sigma(\cdot)$ denotes the nonlinear activation functions, such as $\mathrm{ReLU}(\cdot)$, and $\hat{\mathbf{A}} = \mathbf{A} + \mathbf{I}_n$ stands for the adjacentcy matrix with self-loops, and $\hat{\mathbf{D}}$ is the corresponding degree matrix, and $\mathbf{W}^l$ represents the parameter matrix for the $l$-th layer. Therefore, for the two augmented graphs, their representations can be obtained by computing $\mathbf{H}^{(1)} = \mathrm{GCN}^2(\mathcal{G}_1(\mathbf{A}^{(1)}, \mathbf{X}^{(1)}))$ and $\mathbf{H}^{(2)} = \mathrm{GCN}^2(\mathcal{G}_2(\mathbf{A}^{(2)}, \mathbf{X}^{(2)}))$.

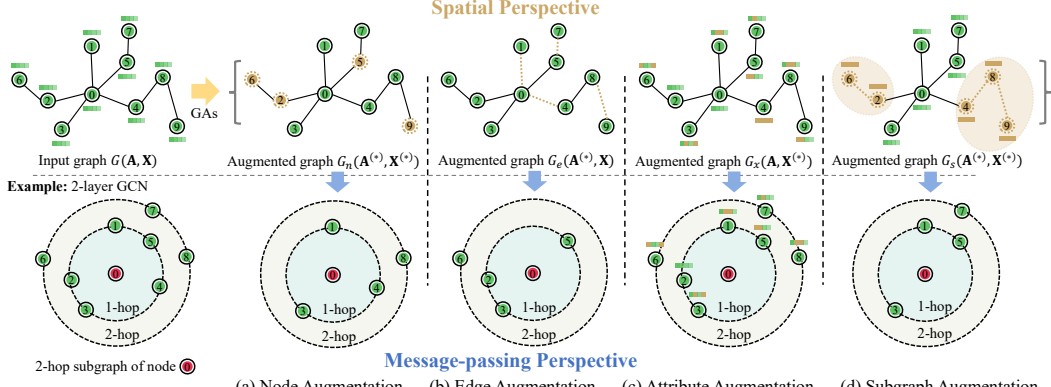

(a) Node Augmentation.  (b) Edge Augmentation.  (c) Attribute Augmentation.  (d) Subgraph Augmentation.

Figure 1: Motivation to unify Graph Augmentations (GAs). A two-hop subgraph example, where the target node is highlighted in red, and the perturbed information is marked in brown. (a) Node augmentation by dropping nodes. (b) Edge augmentation by removing edges. (c) Attribute augmentation by masking attributes. (d) Subgraph augmentation by cropping subgraphs. Existing GAs, typically seen as various forms of global augmentations from the spatial perspective, can be uniformly interpreted as local attribute modifications (*i.e.*, local augmentations) from the message-passing perspective.

**Contrastive losses.** In line with the InfoMax principle [22], various contrastive losses are incorporated in GCLs, guiding the training of graph encoders by maximizing the Mutual Information (MI) between the encoded representations on two augmented graphs. Specifically, given two representations $\mathbf{H}^{(1)}$ and $\mathbf{H}^{(2)}$ obtained from a shared encoder $g_\Theta$, the general objective of GCL is expressed as

$$\text{GCL:} \quad \arg\max_\Theta I(\mathbf{H}^{(1)}; \mathbf{H}^{(2)}), \text{ where } \mathbf{H}^{(1)} = g_\Theta(\mathcal{G}_1), \mathbf{H}^{(2)} = g_\Theta(\mathcal{G}_2), \tag{3}$$

where $I(X; Y)$ represents the MI between $X$ and $Y$. In general, the MI can be approximated using a lower bound estimator, *i.e.*, the InfoNCE loss [33], in the GCLs [53, 20]. This loss can be classified as a sample-level loss because it operates on the sample dimension of the representation matrix. In contrast, Barlow Twins loss [49], another widely employed loss, is designed to remove redundancies among features and hence can be categorized as a feature-level loss. Both losses are used to implement the proposed framework. Section C provides detailed descriptions of these losses.

## 3 Methodology

### 3.1 Motivations

As previously mentioned, Contrastive Learning (CL) seeks to learn image representations invariant to augmentations by encouraging the agreement between embedding vectors from the different image distortions. Due to the employment of identical loss functions, typical Graph Contrastive Learning (GCL) inherits the above representation capabilities from CL. Nonetheless, GCLs should emphasize the *local invariance* owing to the application of graph encoders.

The essence of the graph encoder is to explore the locality of graphs. To be specific, graph encoders in GCLs (generally a 2-layer GCN) follow the message-passing paradigm where node representations are updated in a local aggregation and combination manner, as detailed in Section 2. Given the localizing property of the graph encoder, GAs (*i.e.*, node, edge, attribute, and subgraph augmentations), which are typically viewed as global operations in various forms, can be uniformly reinterpreted as attribute modifications in the neighborhoods of nodes, namely, *local augmentations*, as illustrated in Fig. 1.

1) Edge augmentation involves adding and removing perturbed edges in graphs, equivalent to inserting or masking the attributes of nodes connected by these edges in the neighborhood of impacted nodes. For example, the shown edge removing results in the complete attribute masking of partial 2-hop neighbors (nodes 1, 4, 7, and 8) of the target node 0 during message passing. 2) Attribute augmentation essentially replaces the attributes of perturbed nodes in the graph with new ones, which can be viewed as perturbing the neighborhoods of nodes that contain these perturbed nodes. The attribute masking example shows that during the aggregation phase, the attributes of neighbors (nodes 1, 3, 4, 5, 6, 7,

and 8) on the 2-hop computation graphs of node 0 are masked. 3) Subgraph augmentation is to modify the specific subsets of the graph (including its edges and attributes), which also can be seen as the attribute perturbation in the neighborhoods of target nodes. For node 0, the shown subgraph cropping causes removing nodes 2, 4, 6, and 8 from the 2-hop neighborhood during the message-passing phase. Note that node augmentation is a specific case of subgraph augmentation, where the subset size is one. Thus, the above conclusion regarding subgraph augmentation applies to it.

In short, *the mechanism of GAs in GCLs is to induce attribute modification in the neighborhoods of nodes*. Thus, the essence of GCLs is to learn representations invariant to such local augmentations.

## 3.2 Unified Graph Augmentation Module

Motivated by the above insights, a unified graph augmentation module dubbed UGA is introduced to implement augmentation efficiently and flexibly. The primary idea is to introduce a collection of Augmentation-Centric (AC) vectors for nodes to simulate and exert the impact of GAs on these nodes, namely, attribute variations in the neighborhood of these nodes. A straightforward implementation of UGA is to align AC vectors one-to-one with nodes, match the size of their features, and then perform feature summation to achieve the desired augmentation.

Given the input graph $\mathcal{G}$, the above implementation can be formulated as

$$\text{UGA:} \quad \mathcal{G}_* = \text{t}(\mathcal{G}, \mathbf{Q}), \text{ where } \text{t}(\mathcal{G}, \mathbf{Q}) : \mathcal{G}(\mathbf{A}, \mathbf{X}) \rightarrow \mathcal{G}(\mathbf{A}, \mathbf{X} + \mathbf{Q}), \tag{4}$$

where $\mathcal{G}_*$ denotes an augmented graph derived from the UGA funtion $\text{t}(\mathcal{G}, \mathbf{Q})$, and $\mathbf{Q} \in \mathbb{R}^{n \times f}$ terms the matrix of AC vectors $\mathbf{q}_v \in \mathbb{R}^{1 \times f}$, *i.e.*, AC matrix, and $f$ is the dimension of node attributes.

Fig. 2(a) provides an illustrative example and explains the equivalence between the proposed UGA module and an explicit GA (edge removing). From this, it can be concluded that the UGA module can effectively substitute GAs as long as the combined AC vectors are the representations of cumulative attribute variations within the neighborhoods of nodes induced by these GAs.

**Theorem 3.1.** *Assuming any augmented graph $\mathcal{G}_*(\mathbf{A}^{(*)}, \mathbf{X}^{(*)})$, where $\mathbf{A}^{(*)} \in \mathbb{A}$ and $\mathbf{X}^{(*)} \in \mathbb{X}$, with $\mathbb{A}$ and $\mathbb{X}$ represent the candidate spaces for the augmented adjacency matrix and attribute matrix, respectively, in the proposed implementation of UGA (Eq. 4), there exists an AC matrix $\mathbf{Q}$ that meets*

$$\text{g}_\Theta(\mathbf{A}, \mathbf{X} + \mathbf{Q}) = \text{g}_\Theta(\mathbf{A}^{(*)}, \mathbf{X}^{(*)}), \tag{5}$$

*where $\text{g}_\Theta$ stands for the graph encoder.*

This theorem suggests that the proposed UGA module can be equivalent to any existing GA (including node, edge, attribute, and subgraph operations), thereby demonstrating its *unifying capability* to GAs. Proofs for this theorem are presented in Section D.1. Furthermore, the UGA module possesses an attractive characteristic: *adaptability*, since the AC vectors are capable of dynamically capturing task-relevant perturbation information throughout the training process. Nonetheless, this implementation introduces numerous parameters proportional to the network size, resulting in a significant increase in complexity and the risk of overfitting. To address this limitation, the proposed UGA is reimplemented in a graph-structured manner, where a modest parameter set is utilized, as shown in Fig. 2(b).

**Shared AC vectors.** In graphs, long-range dependencies signify the beyond-local interactions among nodes, represented by similar node attributes and neighborhood patterns [23, 40, 46]. Therefore, it is reasonable to assume that a group of interdependent nodes would benefit from the same optimal GAs. Thus, a shared AC matrix $\mathbf{Q} = [\mathbf{q}_{i,:}]_{i=0}^{k-1}$ is introduced, where $k \ll n$.

**Propagation mode.** AC vectors propagate their features to nodes via a general attention mechanism, in which structural features (*e.g.*, positional/structure encodings [1, 7]) are employed to calculate the attention scores. Formally, the proposed UGA module can be reformulated as

$$\hat{\mathbf{x}}_{v,:} = \mathbf{x}_{v,:} + \sum_{i=0}^{k-1} b_{v,i} \times \mathbf{q}_{i,:}, \quad b_{v,i} = \frac{\exp\left(\text{f}([\mathbf{x}_{v,:}||\mathbf{e}_{v,:}]) \cdot \mathbf{q}_{i,:}^\top\right)}{\sum_{t=0}^{k} \exp\left(\text{f}([\mathbf{x}_{v,:}||\mathbf{e}_{v,:}]) \cdot \mathbf{q}_{t,:}^\top\right)}, \tag{6}$$

where $b_{v,i}$ stands for the propagation weight from $i$-th AC vector to node $v$ within matrix $\mathbf{B} \in \mathbb{R}^{n \times k}$, and $\text{f}([\mathbf{x}_v||\mathbf{e}_v]) \in \mathbb{R}^{1 \times f}$ terms an integrated representation of node $v$, which concatenates the node attributes $\mathbf{x}_v$ and structural features $\mathbf{e}_v \in \mathbb{R}^t$. This paper adopts $t$-steps random-walk encodings [7] as the structure features. $\text{f}(\cdot) : \mathbb{R}^{f+t} \rightarrow \mathbb{R}^f$ denotes a projection layer.

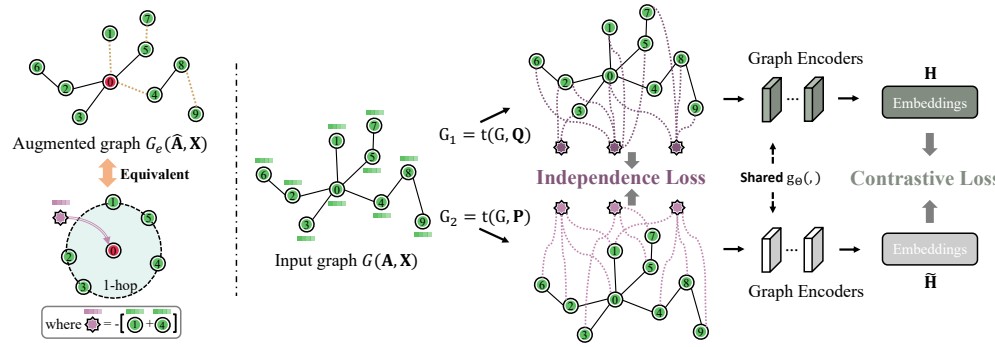

(a) The proposed GA module UGA.    (b) The proposed GCL framework GOUDA.

Figure 2: Illustration of the proposed unified module UGA and generalized framework GOUDA. (a) An intuitive example of the equivalence between GAs (*e.g.*, edge removing) and the aggregation of Augmentation-Centric (AC) vectors, which capture the local attribute variations caused by these GAs. (b) The proposed generalized GCL framework GOUDA. The independence loss, which is directly computed on AC vectors, is designed to ensure diversity across different augmentations.

**Processing.** Keeping a subset of the salient connections facilitates propagation and enhances computational efficiency. Thus, propagation weights below a predefined threshold are zeroed out, namely

$$b_{v,i} = \begin{cases} b_{v,i}, & b_{v,i} > \varepsilon \\ 0, & \text{otherwise}, \end{cases} \tag{7}$$

where $\varepsilon$ terms a threshold value. Then, the obtained weight $\mathbf{B}$ is applied in propagation, per Eq.6.

## 3.3 Generalized Graph Contrastive Learning Framework

Building upon the UGA module, a novel GCL framework named Graph cOntrastive UnifieD Augmentations (GOUDA) is proposed to achieve generality across diverse tasks and graphs. It utilizes the standard two-channel architecture (in Eq. 3), with each channel generating an augmented graph through the UGA module and its AC matrix, as depicted in Fig. 2. Unlike traditional GCLs, GOUDA introduces a term to constrain the two AC matrices (denoted as $\mathbf{Q}$ and $\mathbf{P}$).

Specifically, GOUDA optimizes the following objective function:

$$\text{GOUDA:} \quad \underset{\Theta}{\text{argmax}} \; I\left(g_\Theta\left(\mathcal{G}_1\right); g_\Theta\left(\mathcal{G}_2\right)\right) + \mathcal{D}\left(\mathbf{Q}, \mathbf{P}\right), \tag{8}$$

where $I(X;Y)$ stands for the Mutual Information (MI) between $X$ and $Y$, and $g_\Theta$ denotes a graph encoder shared between two views (or channels). $\mathcal{G}_1 = t\left(\mathcal{G}, \mathbf{Q}\right)$ and $\mathcal{G}_2 = t\left(\mathcal{G}, \mathbf{P}\right)$ represent two augmented graphs, and $\mathcal{D}\left(\mathbf{Q}, \mathbf{P}\right)$ denotes the constraint between $\mathbf{Q}$ and $\mathbf{P}$.

**Definition 3.2.** (Consistency across augmentations). Let $\mathbf{H}^{(i)}$ and $\mathbf{H}^{(j)}$ denote the representations of graphs $\mathcal{G}_i, \mathcal{G}_j \sim \mathbb{G}_\omega$, respectively, where $j \neq i$, and $\mathbb{G}_\omega$ denotes the family of graphs derived from a series of parametric graph augmentations. Consistency across augmentations for node $v$ is defined as

$$\mathcal{C}_v = \mathcal{S}\left(\mathbf{h}_v^{(i)}, \mathbf{h}_v^{(j)}\right), \tag{9}$$

where $\mathcal{S}(X, Y)$ terms the distributional similarity between $X$ and $Y$.

This consistency implies that augmentation should minimally impact the similarity between representations from different augmented graphs for the same nodes to preserve the intrinsic semantic integrity of the nodes. Note that the first term in the objective function of GOUDA, namely the MI maximization, essentially is a constraint for semantic consistency. Thus, the augmentation learned by the UGA module is capable of ensuring the desired property.

**Definition 3.3.** (Diversity across augmentations). Given two augmented graphs $\mathcal{G}_i(\mathbf{A}^{(i)}, \mathbf{X}^{(i)})$ and $\mathcal{G}_j(\mathbf{A}^{(j)}, \mathbf{X}^{(j)}) \sim \mathbb{G}_\omega$, where $j \neq i$, let $\mathcal{N}_v^k$ represents the $k$-hop subgraph centered at node $v$, and let $\mathcal{D}(,)$ stands for the measure of distributional difference. Diversity across augmentations is defined as

$$\mathcal{D}_v = \mathcal{D}\left(\text{COM}(\mathbf{X}_{\mathcal{N}_v^k}^{(i)}), \text{COM}(\mathbf{X}_{\mathcal{N}_v^k}^{(j)})\right), \tag{10}$$

where $\mathbf{X}_{\mathcal{N}_v^k}^{(i)} \in \mathbb{R}^{n_v \times f}$ stands for the attribute matrix of nodes in the $k$-hop subgraph of node $v$, and $n_v$ is the number of neighbors for node $v$. $\mathrm{COM}(\cdot)$ terms a combination function, such as $\mathrm{sum}(\cdot)$.

This definition is based on the conclusion in Section 3.1, namely, the mechanism of GAs in GCLs is to modify attributes within the node neighborhoods. Accordingly, another objective for augmentation is the minimization of the local attribute overlap between augmented graphs, ensuring the model does not overfocus on the specific features of a single distribution.

In the proposed GOUDA framework, local attribute variations for each node are represented by AC vectors, $e.g.$, $\mathbf{q}_v$ and $\mathbf{p}_v$, with the augmentation being generated by aggregating features from these vectors. Therefore, the diversity across augmentations can be quantified using two AC matrices ($\mathbf{Q}$ and $\mathbf{P}$) corresponding to two distinct views, which is supported by the following analysis.

**Theorem 3.4.** *Let $\mathcal{D}(X, Y) = \|X - Y\|_F^2$ stands for the distributional difference, and let $\mathrm{COM}(\cdot) = \mathrm{sum}(\cdot)$ terms the combination function. In GOUDA (in Eq. 8), the diversity across augmentations (in Eq. 10) can be approximated by the distributional difference between AC matrices $\mathbf{Q}$ and $\mathbf{P}$, that is*

$$\mathcal{D}_v = \| \sum_{t \in \mathcal{N}_v \cup v} (\mathbf{x}_{t,:}^{(1)} + \hat{\mathbf{q}}_{t,:}) - \sum_{t \in \mathcal{N}_v \cup v} (\mathbf{x}_{t,:}^{(2)} + \hat{\mathbf{p}}_{t,:}) \|_F^2 \approx \|\mathbf{Q} - \mathbf{P}\|_F^2, \quad (11)$$

*where $\hat{\mathbf{q}}_{t,:} = \mathbf{b}_{t,:}^q \mathbf{Q}$ denote the features propagated from AC matrices $\mathbf{Q}$ to node $t$.*

Theorem 3.4 shows that the diversity across augmentations can be controlled by imposing constraints on the AC matrices, particularly through the second term in the objective function of GOUDA. Refer to Section D.2 for the proofs. In brief, maintaining a balance between consistency and diversity across augmentations is crucial for the effectiveness of GCLs. Specifically, diversity encourages exploring and exploiting the local attribute variations while consistency anchors the learned representations to the original semantics.

## 3.4 Instantiation of GOUDA

This subsection introduces a practical implementation of the proposed GOUDA framework (Eq.8). The overview of this framework is depicted in Fig. 2, while the step-by-step procedure is detailed in Algorithm 1. The objective of GOUDA is to learn the discriminative and robust representations. To achieve this, it seeks to train the graph encoder $g_\Theta$ to maximize the Mutual Information (MI) between representations from two augmented graphs $\mathcal{G}_1 = \mathrm{t}(\mathcal{G}, \mathbf{Q})$ and $\mathcal{G}_2 = \mathrm{t}(\mathcal{G}, \mathbf{P})$, while simultaneously maintaining consistency and diversity in the augmentation process.

**Estimation of Mutual Information (MI).** The first term of GOUDA is implemented utilizing the sample-level InfoNCE loss (in Eq. 22), which serves as a lower bound estimator for MI, and the feature-level Barlow Twins loss (in Eq. 24). This term is denoted as contrastive loss $\mathcal{L}_{\mathrm{contrast}}$. Owing to limited space, the above losses are introduced in Section C.

**Distributional independence loss.** A distributional independence loss is introduced to instantiate the second term of GOUDA. Specifically, the Hilbert-Schmidt Independence Criterion (HSIC) [11] is adopted to measure the statistical dependence between two augmentation distributions. Furthermore, the Gram matrices derived from HSIC are constrained to minimize their off-diagonal elements. To be concrete, the independence loss is formulated as

$$\mathcal{L}_{\mathrm{indep}} = \underbrace{1/(n-1)^2 \, \mathrm{trace}\,(\mathbf{KRLR})}_{\mathrm{HSIC}} + \beta_1 \sum_i \sum_{j \neq i} k_{i,j} + \beta_2 \sum_i \sum_{j \neq i} l_{i,j}, \quad (12)$$

where $\mathbf{K}$ and $\mathbf{L}$ stand for the Gram matrices of $\mathbf{Q}$ and $\mathbf{P}$, respectively, defined by $k_{i,j} = \kappa(\mathbf{q}_{i,:}, \mathbf{q}_{j,:})$ and $l_{i,j} = \kappa(\mathbf{p}_{i,:}, \mathbf{p}_{j,:})$. In practice, the kernel function $\kappa(\cdot)$ is defined as the linear kernel, specifically $k_{i,j} = \mathbf{q}_{i,:} \mathbf{q}_{j,:}^T$. Additionally, $\mathbf{R} = \mathbf{I}_n - \frac{1}{n}\mathbf{1}\mathbf{1}^\top$ represents the centering matrix, where $\mathbf{I} \in \mathbb{R}^{n \times n}$ and $\mathbf{1}_n \in \mathbb{R}^{n \times 1}$ denote the identity matrix and all-one column vector, respectively. $\beta_1$ and $\beta_2$ represent two hyper-parameters. Minimizing this term serves two purposes: on the one hand, it enhances the diversity across augmentations by amplifying the differences between two distributions, and on the other hand, it avoids trivial solutions by increasing the differences among the augmentation elements ($\mathbf{q}_{i,:}$) within each distribution.

**Objective.** The overall objective function of GOUDA is a weighted sum of these two terms, that is

$$\mathcal{L} = \mathcal{L}_{\mathrm{contrast}} + \gamma \mathcal{L}_{\mathrm{indep}}, \quad (13)$$

where $\gamma$ denotes a hyperparameter used to trade off two terms.

Table 1: Comparison of time complexity in the *augmentation* Phase. $n$ denotes the size of the graph.

| Model | Time Complexity | Description |
|---|---|---|
| SPAN [21] | $O(n^2 tk)$ | Eigendecomposition-based edge augmentation. |
| JOAO [47] | $O(n^2 d)$ | Min-max optimization-based augmentation. |
| AD-GCL [31] | $O(n^2 d)$ | Adversarial-training-based edge augmentation. |
| GOUDA (Ours) | $O(nkf)$ | Consistency-diversity balanced augmentation. |

Table 2: Accuracy in percentage (mean$_{\pm \text{std}}$) over ten trials of node classification across seven graphs. Best and runner-up models are highlighted in **bolded** and underlined, respectively.

| Model | Input | Cora | CiteSeer | PubMed | Wiki-CS | Photo | Computers | Physics |
|---|---|---|---|---|---|---|---|---|
| GCN | A, X, Y | $82.32_{\pm 1.79}$ | $72.13_{\pm 1.17}$ | $84.90_{\pm 0.38}$ | $76.89_{\pm 0.37}$ | $92.35_{\pm 0.25}$ | $86.34_{\pm 0.48}$ | $95.65_{\pm 0.16}$ |
| GAT | A, X, Y | $83.34_{\pm 1.57}$ | $72.44_{\pm 1.42}$ | $85.21_{\pm 0.36}$ | $77.42_{\pm 0.19}$ | $92.35_{\pm 0.25}$ | $87.06_{\pm 0.35}$ | $95.47_{\pm 0.15}$ |
| DGI | A, X | $82.60_{\pm 0.40}$ | $71.49_{\pm 0.14}$ | $86.00_{\pm 0.14}$ | $75.73_{\pm 0.13}$ | $91.49_{\pm 0.25}$ | $84.09_{\pm 0.39}$ | $94.51_{\pm 0.52}$ |
| GMI | A, X | $82.51_{\pm 1.47}$ | $71.56_{\pm 0.56}$ | $84.83_{\pm 0.90}$ | $75.06_{\pm 0.13}$ | $90.72_{\pm 0.33}$ | $81.76_{\pm 0.52}$ | $94.10_{\pm 0.61}$ |
| MVGRL | A, X | $83.03_{\pm 0.27}$ | $72.75_{\pm 0.46}$ | $85.63_{\pm 0.38}$ | $77.97_{\pm 0.18}$ | $92.01_{\pm 0.13}$ | $87.09_{\pm 0.27}$ | $95.33_{\pm 0.03}$ |
| GRACE | A, X | $83.30_{\pm 0.40}$ | $71.41_{\pm 0.38}$ | $86.51_{\pm 0.34}$ | $79.16_{\pm 0.36}$ | $92.65_{\pm 0.32}$ | $87.21_{\pm 0.44}$ | $95.26_{\pm 0.02}$ |
| GCA | A, X | $83.90_{\pm 0.41}$ | $72.21_{\pm 0.24}$ | $86.01_{\pm 0.75}$ | $79.35_{\pm 0.12}$ | $92.78_{\pm 0.17}$ | $87.84_{\pm 0.27}$ | $95.68_{\pm 0.05}$ |
| BGRL | A, X | $83.77_{\pm 0.75}$ | $71.99_{\pm 0.42}$ | $84.94_{\pm 0.17}$ | $78.74_{\pm 0.22}$ | $93.24_{\pm 0.29}$ | $88.92_{\pm 0.33}$ | $95.63_{\pm 0.04}$ |
| GBT | A, X | $83.89_{\pm 0.66}$ | $72.57_{\pm 0.61}$ | $85.71_{\pm 0.32}$ | $76.65_{\pm 0.62}$ | $92.63_{\pm 0.44}$ | $88.14_{\pm 0.33}$ | $95.07_{\pm 0.17}$ |
| CCA-SSG | A, X | $84.39_{\pm 0.68}$ | $73.81_{\pm 0.38}$ | $86.21_{\pm 0.67}$ | $78.94_{\pm 0.17}$ | $93.14_{\pm 0.14}$ | $88.74_{\pm 0.28}$ | $95.38_{\pm 0.06}$ |
| SPAN | A, X | $85.09_{\pm 0.28}$ | $73.68_{\pm 0.53}$ | $85.35_{\pm 0.29}$ | $79.01_{\pm 0.51}$ | $92.68_{\pm 0.31}$ | $\textbf{89.68}_{\pm 0.19}$ | $95.12_{\pm 0.15}$ |
| DSSL | A, X | $84.52_{\pm 0.71}$ | $73.93_{\pm 0.89}$ | $85.59_{\pm 0.28}$ | $79.98_{\pm 0.67}$ | $93.08_{\pm 0.38}$ | $89.06_{\pm 0.49}$ | $95.29_{\pm 0.29}$ |
| HomoGCL | A, X | $84.89_{\pm 0.71}$ | $73.78_{\pm 0.63}$ | $86.37_{\pm 0.49}$ | $79.29_{\pm 0.32}$ | $92.92_{\pm 0.18}$ | $88.46_{\pm 0.20}$ | $95.18_{\pm 0.09}$ |
| GOUDA-IF | A, X | $\textbf{86.11}_{\pm 0.55}$ | $\textbf{74.55}_{\pm 0.97}$ | $87.55_{\pm 0.10}$ | $\textbf{80.61}_{\pm 0.28}$ | $93.69_{\pm 0.32}$ | $89.21_{\pm 0.17}$ | $96.09_{\pm 0.14}$ |
| GOUDA-BT | A, X | $85.99_{\pm 0.31}$ | $74.47_{\pm 1.05}$ | $\textbf{87.59}_{\pm 0.02}$ | $80.37_{\pm 0.30}$ | $\textbf{93.82}_{\pm 0.19}$ | $89.55_{\pm 0.11}$ | $\textbf{96.19}_{\pm 0.21}$ |

## 3.5 Complexity Analysis

This subsection evaluates the complexity of the proposed GOUDA framework in comparison to the baseline GCLs configured with learnable GAs, including SPAN, JOAO, and AD-GCL. As illustrated in Tab. 1, GOUDA introduces lighter computational overhead compared to these baselines. For a detailed description of the complexity, refer to Section E.4.

## 4 Experiments

This section evaluates the effectiveness and generality of the proposed GOUDA through a comprehensive comparison against multiple baselines across tasks at both the node-level (node classification and node clustering) and the graph-level (graph classification) tasks. Furthermore, it conducts several additional experiments to deepen the understanding of this framework. For an exhaustive account of datasets, baselines, configurations, and hyper-parameters, refer to Section E.

**Datasets.** The experiment utilizes ten benchmark datasets, namely: Cora [28], CiteSeer [28], PubMed [28], Wiki-CS [25], Photo [29], Computers [29], and Physics [29] for node-level tasks, and IMDB-B [42], IMDB-M [42], and COLLAB [42] for graph-level tasks. See Section E.1 for dataset descriptions.

**Baselines.** The baseline models comprise two supervised graph neural networks (GCN [17], GAT [34]) and eleven self-supervised graph learning models (DGI [35], GMI [26], MVGRL [14], GRACE [53], GCA [54], BGRL [32], CCA-SSG [50], GBT [2], SPAN [21], DSSL [39], HomoGCL [20]) for node-level tasks. Four self-supervised learning models are compared (InfoGraph [30], GraphCL [48], JOAO [47], AD-GCL [31]) for graph-level tasks. Refer to Section E.3 for model introductions.

### 4.1 Experimental Results

**Node Classification.** It can be observed from Tab. 2, which exhibits the results of node classification tasks, that the proposed GOUDA outperforms the unsupervised baselines in six of the seven datasets. This demonstrates the superiority of GOUDA. Furthermore, on the CiteSeer dataset, notable performance improvements are observed with both models, GOUDA-IF and GOUDA-BT, surpassing

the baselines GRACE and GBT. Specifically, the accuracy of GOUDA-IF surpasses that of GRACE by $3.14\%$, and similarly, the accuracy of GOUDA-BT exceeds that of GBT by $1.90\%$. Note that the baselines GRACE and GBT adopt identical encoders and contrastive losses as GOUDA-IF and GOUDA-BT, respectively. Therefore, the observed performance improvement can be attributed to the adaptive modeling capacity for augmentations of the proposed GOUDA.

**Node Clustering.** One can draw two conclusions from the observations in Tab. 3. Firstly, it is evident that GOUDA consistently surpasses all baselines (*e.g.*, GRACE, and GBT) across all datasets, which illustrates the superior representation capacity of GOUDA. This can be attributed to the self-adaptive learning ability of the proposed module UGA. Secondly, GOUDA-IF consistently outperforms GOUDA-BT, suggesting that within the proposed GOUDA framework, the InfoNCE loss more effectively captures local information for clustering than the BarlowTwins loss.

Table 3: Performances on node clustering: NMI & ARI Scores in percentage (mean).

| | Cora | | CiteSeer | | PubMed | |
|---|---|---|---|---|---|---|
| | NMI | ARI | NMI | ARI | NMI | ARI |
| DGI | 52.75 | 47.78 | 40.43 | 41.84 | 30.03 | 29.78 |
| MVGRL | 54.21 | 49.04 | 43.26 | 42.73 | 30.75 | 30.42 |
| GRACE | 54.59 | 48.31 | 43.02 | 42.32 | 31.11 | 30.37 |
| GBT | 55.32 | 48.91 | 44.01 | 42.61 | 31.33 | 30.64 |
| CCA-SSG | 56.38 | 50.62 | 43.98 | 42.79 | 32.06 | 31.15 |
| GOUDA-IF | **57.92** | **52.41** | **45.11** | **43.82** | **33.17** | **31.98** |
| GOUDA-BT | 57.35 | 51.84 | 44.93 | 43.46 | 33.14 | 31.73 |

**Graph Classification.** The results of this experiment are presented in Tab. 4 and Fig. 3. Firstly, it can be observed from Tab. 4 that GOUDA outperforms the baselines regarding classification performance, which illustrates the general validity of GOUDA. In particular, GOUDA-IF and GOUDA-BT surpass the second-place MVGRL by $1.02\%$ and $2.6\%$, respectively, on the IMDB-B dataset, which highlights the superiority of GOUDA. Moreover, GOUDA exceeds the GCLs employing learnable GAs, *i.e.*, JOAO, and AD-GCL. This can be due to the unified ability of UGA to integrate diverse GAs, which provides GOUDA with broader augmentation options than the baseline. Secondly, as illustrated in Fig. 3, GOUDA achieves superior performance and consumes less time than the baselines. Specifically, two triangles, indicating the proposed GOUDA, are superior left and above the other shape in the figure. This implies GOUDA is lightweight, aligning with conclusions in Section 3.5. Besides, UGA introduces modest memory usage, which promises the scalability of GOUDA.

Table 4: Performances on graph classification: accuracy in percentage (mean$_{\pm \text{std}}$).

| Model | IMDB-B | IMDB-M | COLLAB |
|---|---|---|---|
| Infograph | $73.03_{\pm 0.87}$ | $49.69_{\pm 0.53}$ | $82.00_{\pm 0.29}$ |
| GraphCL | $71.14_{\pm 0.44}$ | $48.58_{\pm 0.67}$ | $71.36_{\pm 1.15}$ |
| JOAO | $71.60_{\pm 0.86}$ | $49.20_{\pm 0.77}$ | $70.40_{\pm 2.21}$ |
| AD-GCL | $71.49_{\pm 0.90}$ | $50.36_{\pm 0.74}$ | $74.89_{\pm 0.90}$ |
| MVGRL | $74.20_{\pm 0.70}$ | $51.20_{\pm 0.50}$ | $73.10_{\pm 0.60}$ |
| GOUDA-IF | $75.22_{\pm 0.94}$ | $52.43_{\pm 0.83}$ | $\mathbf{85.70}_{\pm 2.33}$ |
| GOUDA-BT | $\mathbf{76.80}_{\pm 0.98}$ | $\mathbf{53.05}_{\pm 0.72}$ | $85.15_{\pm 2.17}$ |

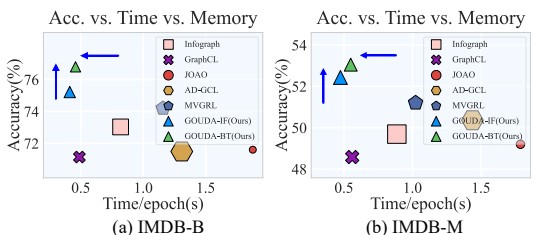

Figure 3: Comparisons in terms of performance, running time, and GPU memory usage. The marker size indicates memory usage.

## 4.2 Additional Experiments

**Robustness Analysis.** This experiment aims to evaluate the robustness of GOUDA against topology attacks (adding edges) and attribute attacks (flipping attributes). According to results in Fig. 4 and Fig. 5, several conclusions can be derived. Firstly, compared to the baselines using the same contrastive losses, GOUDA consistently achieves performance gains at all perturbation rates. It demonstrates the robustness of GOUDA against both topology and attribute attacks. This is attributed to the greater adaptability of the UGA module, stemming from its integration of augmentation and contrastive updating over random GAs. Secondly, attribute attacks cause more severe performance degradation than topology attacks, even for our proposed GOUDA. This could be because the node attributes, rich with class-discriminative information, are erased essential identification info during attacks.

**Ablation Study.** This experiment aims to evaluate the contribution of individual components. To be specific, it introduces two variants: one without the structure features (in Eq. 6) and another without the independence loss (in Eq. 13). From Fig. 6, it is observable that the performance declined in both model variants compared to the complete model, which illustrates that the efficacy of GOUDA stems from the collective contribution of all components. Besides lacking Independence loss, GOUDA-IF performs inferior to GOUDA-BT, implying that InfoNCE might drive the model toward excessive

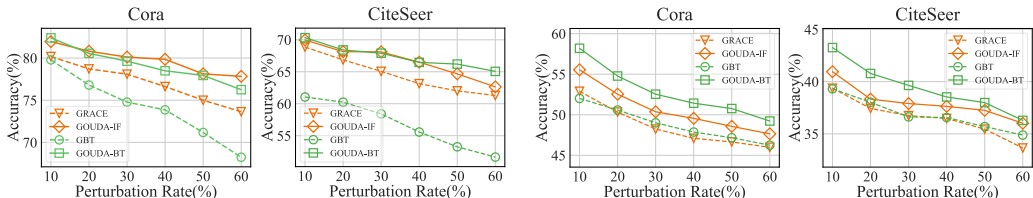

Figure 4: Topology attack effects on GCLs.     Figure 5: Attribute attack effects on GCLs.

consistency, diminishing its discriminative power. This highlights the critical role of independence loss in preserving diversity across augmentations.

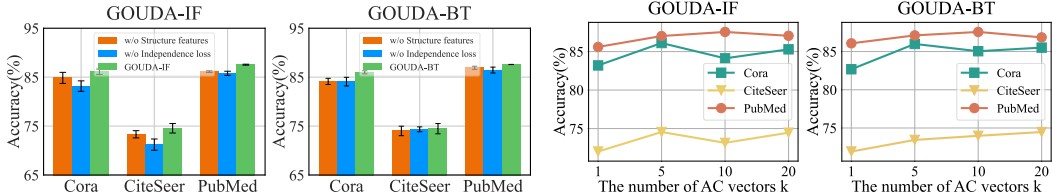

Figure 6: Contribution of individual components.     Figure 7: Impact of the number of AC vectors.

**Parameter Sensitivity Analysis.** These experiments are performed to offer an intuitive understanding of hyper-parameter selection. Firstly, as depicted in Fig. 7, which illustrates performance variance for varying $k$, GOUDA achieves consistently stable performances across $\{5, 10, 20\}$. Notably, the performance variation on these datasets remains minimal, staying within a $2\%$ margin. Thus, GOUDA has low sensitivity to parameter $k$. Moreover, this parameter requires no significant value for GOUDA to perform well; a setting as low as $5$ suffices. However, a value of $1$ is inadequate due to the absence of augmentation diversity. The analysis of other hyper-parameters is given in Section E.6.

## 5   Conclusions

In this paper, we present UGA, a unified Graph Augmentation (GA) module that addresses the issues in existing GAs, including specificity, complexity, and incompleteness. Motivated by the local attribute-modifying characteristics of GAs, UGA introduces a moderate number of Augmentation-Centric (AC) vectors to simulate GA impact on nodes. We further propose GOUDA, a generalized Graph Contrastive Learning (GCL) framework built on UGA. GOUDA promotes both consistency and diversity across augmentations by employing a contrastive loss and an independence loss, respectively. Extensive evaluations demonstrate the generality and efficiency of GOUDA. However, the robustness analysis suggests a scope for enhancement in its robustness against attribute attacks. Future research could explore multi-modal learning methods that fuse diverse structural features into node attributes, aiming to better preserve discriminative information and thus enhance robustness.

## 6   Acknowledgments

This work was supported in part by the National Key R&D Program of China (No. 2022ZD0119202), in part by the National Natural Science Foundation of China (No. U22B2036, 62376088, 62276187, 62102413, 62272020), in part by the Hebei Natural Science Foundation (No. F2024202047), in part by the Hebei Province Higher Education Science and Technology Research Project (No. QN2024201), in part by the National Science Fund for Distinguished Young Scholarship (No. 62025602), and in part by the XPLORER PRIZE.

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

# A  Algorithm Description

To demonstrate the broad applicability of the proposed framework GOUDA, this paper implements two models, namely GOUDA-IF and GOUDA-BT. Specifically, GOUDA-IF employs a prevalent node-level contrastive loss ( *i.e.*, InfoNCE), while GOUDA-BT utilizes a feature-level contrastive loss (*i.e.*, BarlowTwins). GOUDA-IF is illustrated as an example to describe the entire algorithm, as presented in Algorithm 1. Accordingly, GOUDA-BT can be described by substituting the contrastive loss $\mathcal{L}_{contrast}(,)$ in line 3 of this algorithm.

---

**Algorithm 1:** GOUDA-IF

---

**Input:**  graph $\mathcal{G}(\mathbf{A}, \mathbf{X})$, hyperparameters $\gamma$, $\beta_1$, $\beta_2$ and $\tau$.
**Output:**  node representations $\mathbf{H}^{(1)} \in \mathbb{R}^{n \times d}$ and $\mathbf{H}^{(2)} \in \mathbb{R}^{n \times d}$.
**Initialization:** graph encoder $g_\Theta(,)$, projection heads $g(\cdot)$, and the matrices of
  Augmentation-Centric (AC) vectors $\mathbf{Q} \in \mathbb{R}^{k \times f}$ and $\mathbf{P} \in \mathbb{R}^{k \times f}$.

**while** not converged **do**
    % Augmentation %
    1. $\mathcal{G}_1 \leftarrow t(\mathcal{G}, \mathbf{Q})$ and $\mathcal{G}_2 \leftarrow t(\mathcal{G}, \mathbf{P})$ via Eq. 6 and Eq. 7;
    % Encoding %
    2. $\mathbf{H}^{(1)} \leftarrow g_\Theta(\mathcal{G}_1)$ and $\mathbf{H}^{(2)} \leftarrow g_\Theta(\mathcal{G}_2)$ via Eq. 2;
    % Calculating loss %
    3. $\mathcal{L} \leftarrow \mathcal{L}_{\text{contrast}}(\mathbf{H}^{(1)}, \mathbf{H}^{(2)}) + \gamma \mathcal{L}_{\text{indep}}(\mathbf{Q}, \mathbf{P})$ via Eq. 13;
    % Optimizing %
    4. $\Theta \leftarrow \text{Adam}(\mathcal{L}, \Theta)$;
**end**
**return** $\mathbf{H}^{(1)} \in \mathbb{R}^{n \times d}$, $\mathbf{H}^{(2)} \in \mathbb{R}^{n \times d}$ and $g_\Theta(,)$;

---

# B  Introduction of Graph Augmenations

Categorized by the type of graph information they manipulate, Graph Augmentations (GAs) can be broadly divided into four categories: node augmentation, edge augmentation, attribute augmentation, and subgraph augmentation. The detailed introduction and examples are given below.

**Node Augmentation.** This augmentation generally creates the new graph by dropping or adding the perturbated nodes and the edges that connect to these perturbated nodes of the input graph, as shown in Fig. 1(a). Employing the adjacent matrix to formally represent the graph topology, the edge augmentation can be denoted by $\mathcal{G}_n(\mathbf{A}^{(*)}, \mathbf{X}^{(*)})$. Therefore, these can be formulated as

$$\text{Node Dropping}: \quad \mathbf{A}^{(*)}, \mathbf{X}^{(*)} \leftarrow \{\mathcal{V}/\bar{\mathcal{V}}, \mathcal{E}/\bar{\mathcal{E}}\}, \mathbf{X}/\bar{\mathbf{X}} \tag{14}$$

$$\text{Node Adding}: \quad \mathbf{A}^{(*)}, \mathbf{X}^{(*)} \leftarrow \{\mathcal{V} \cup \bar{\mathcal{V}}, \mathcal{E} \cup \bar{\mathcal{E}}\}, \mathbf{X}||\bar{\mathbf{X}}, \tag{15}$$

where $\bar{\mathcal{V}}$ denotes the perturbated node set, $\bar{\mathbf{X}}$ stands for the attributes of these nodes, and $\bar{\mathcal{E}}$ terms the set of edges connected to these nodes. The operator is widely used on graph classification [48, 47].

**Edge Augmentation.** Unlike node augmentation, the edge augmentation, denoted by $\mathcal{G}_e(\mathbf{A}^{(*)}, \mathbf{X})$, exclusively operate on edges. It involves either removing perturbated edges from or adding perturbated edges to the input graph, as indicated in Fig. 1(b). These can be expressed as

$$\text{Edge Removing}: \quad \mathbf{A}^{(*)} \leftarrow \{\mathcal{E}/\bar{\mathcal{E}}\} \tag{16}$$

$$\text{Edge Adding}: \quad \mathbf{A}^{(*)} \leftarrow \{\mathcal{E} \cup \bar{\mathcal{E}}\}, \tag{17}$$

where $\bar{\mathcal{E}}$ represents the set of perturbated edges, which is randomly determined [53, 54, 32, 20] or adaptively learned [31] during each training epoch.

**Attribute Augmentation.** Typically, the attribute augmentation (expressed by $\mathcal{G}_a(\mathbf{A}, \mathbf{X}^{(*)})$) generates the new graph by masking (in Fig. 1(c)) or corrupting the raw node attributes. These can be described as

$$\text{Attribute Masking}: \quad \mathbf{X}^{(*)} = \mathbf{X} \odot \mathbf{M} \tag{18}$$

$$\text{Attribute Corrupting}: \quad \mathbf{x}_v^{(*)} = \mathbf{x}_v + \delta_v, \tag{19}$$

where $\mathbf{M} \in \mathbb{R}^{n \times f}$ stands for the mask matrix and $\delta_v$ terms the noise vector for node $v$, which is updated iteratively by adversarial training [19, 44].

**Subgraph Augmentation.** As typical graph-level operators, the subgraph augmentation crops out subgraphs (in Fig. 1(d)) or inserts additional subgraphs to create new graphs, as follows

$$\text{Subgraph Cropping}: \quad \mathcal{G}_s(\mathbf{A}^{(*)}, \mathbf{X}^{(*)}) \leftarrow \mathcal{G} \cup \bar{\mathcal{G}}(\bar{\mathbf{A}}, \bar{\mathbf{X}}) \tag{20}$$

$$\text{Subgraph Inserting}: \quad \mathcal{G}_s(\mathbf{A}^{(*)}, \mathbf{X}^{(*)}) \leftarrow \mathcal{G}/\bar{\mathcal{G}}(\bar{\mathbf{A}}, \bar{\mathbf{X}}), \tag{21}$$

where $\bar{\mathcal{G}}(\bar{\mathbf{A}}, \bar{\mathbf{X}})$ stands for the perturbated subgraph. The subgraph augmentation is mostly used for graph-level tasks [47, 13].

## C  Introduction of Graph Contrastive Losses

The contrastive loss serves as a crucial technique that enhances data representation through discrimination. Generally, it operates on two levels of representation matrices: the sample level [4, 53], where it aligns the representations of positive samples and uniformly distributes all representations, and the feature level [49, 2], where it targets reducing redundancy between features.

**Sample-level contrastive losses.** In the nascent stages of research, the designs of contrastive losses were inspired by the success of contrastive learning (CL) in computer vision (CV) [4]. To be specific, this type of contrastive loss aims to minimize the distance between the anchor sample and positive samples while maximizing the distance between the anchor sample and negative samples [33]. As a typically sample-level contrastive loss, InfoNCE loss [33] classifies the embeddings of the same node from different views as positive samples while treating the embeddings from all other nodes as negative samples. This loss can be formulated as

$$\mathcal{L}_{\text{InfoNCE}} = \frac{1}{2n} \sum_{v \in \mathcal{V}} \left( \ell(\mathbf{h}_v^{(1)}, \mathbf{h}_v^{(2)}) + \ell(\mathbf{h}_v^{(2)}, \mathbf{h}_v^{(1)}) \right), \tag{22}$$

$$\text{where} \quad \ell\left(\mathbf{h}_v^{(1)}, \mathbf{h}_v^{(2)}\right) = -\log \frac{e^{\Phi\left(\mathbf{h}_v^{(1)}, \mathbf{h}_v^{(2)}\right)/\tau}}{\sum\limits_{u \in \mathcal{V}} e^{\Phi\left(\mathbf{h}_v^{(1)}, \mathbf{h}_u^{(2)}\right)/\tau} + \sum\limits_{u \in \mathcal{V}, u \neq v} e^{\Phi\left(\mathbf{h}_v^{(1)}, \mathbf{h}_u^{(1)}\right)/\tau}}, \tag{23}$$

where $\Phi(\mathbf{h}_v, \mathbf{h}_u) = \mathrm{s}\left(g(\mathbf{h}_v), g(\mathbf{h}_u)\right)$ stands for the feature similarity function, and $g(\cdot)$ represents the projection heads [4], and $\mathrm{s}(\cdot)$ terms the consine similarity. $\tau$ denotes the temperature coefficient.

**Feature-level contrastive losses.** This type of loss is designed to directly optimize contrastive losses in the feature space, bypassing the need for defining explicit positive and negative samples, thus overcoming sample selection challenges. Barlow Twins (BT) [49] and CCA-SSG [50] are two notable approaches aimed at improving feature representation learning by minimizing redundancy between feature dimensions. BT enforces the mutual information matrix computed on the features from two different views to approximate the identity matrix, ensuring that the learned representations are free from redundant information. This can be formulated as

$$\mathcal{L}_{\text{BarlowTwins}} = \sum_{i=0}^{f-1}(1-c_{i,i}) + \lambda \sum_{i=0}^{f-1}\sum_{\substack{j=0\\j\neq i}}^{f-1}(c_{i,j})^2, \quad c_{i,j} = \frac{\sum_{v \in \mathcal{V}} h_{v,i} \times h_{v,j}}{\sqrt{\sum_{v \in \mathcal{V}}(h_{v,i})^2} \times \sqrt{\sum_{v \in \mathcal{V}}(\tilde{h}_{v,j})^2}}, \tag{24}$$

where $\lambda$ denotes a hyperparameter to tradeoff two terms.

CCA-SSG not only pushes the covariance matrices from each view to approach an identity matrix but also enhances feature consistency across views to learn informative representations of both unique and shared data characteristics.

## D  Theoretical analysis

### D.1  Proofs for Theorem 3.1

For the sake of clarity, let us briefly describe the whole process. Firstly, the proof proceeds from the premise of a one-layer graph encoder without activation functions. For node-level tasks, the graph

encoder $g_{\Theta}(,)$ is configured as a one-layer GCN. For graph-level tasks, $g_{\Theta}(,)$ is set as a one-layer GIN with a sum pooling $\text{sum}(\cdot)$. Subsequently, the obtained conclusions are generalized to the case of multi-layer encoders.

Specifically, for the one-layer graph encoders, the encoding process for node representations ($\mathbf{H}$) and graph representations ($\mathbf{h}$) can be expressed as

$$\text{Node representations:} \quad \mathbf{H} = \text{GCN}(\mathbf{A}, \mathbf{X}) = \tilde{\mathbf{A}} \cdot \mathbf{X} \cdot \mathbf{W} \tag{25}$$
$$\text{Graph representations:} \quad \mathbf{h} = \text{sum}(\text{GIN}(\mathbf{A}, \mathbf{X})) = \text{sum}\left((\mathbf{A} + (1 + \epsilon)\mathbf{I}) \cdot \mathbf{X} \cdot \mathbf{W}\right), \tag{26}$$

where $\tilde{\mathbf{A}} = \hat{\mathbf{D}}^{-\frac{1}{2}} \hat{\mathbf{A}} \hat{\mathbf{D}}^{-\frac{1}{2}}$ denotes the normalized adjacency matrix and $\epsilon$ terms an learnable parameter.

In the proposed UGA implementation, both node and graph representations can be decoupled into two terms: the original representations (represented as $\mathbf{H}$ and $\mathbf{h}$), which can be regarded as directly calculated from Eq. 25 and Eq. 26, respectively, and the augmented representations (denoted as $\triangle\mathbf{H}^q$ and $\triangle\mathbf{h}^q$). To be specific, two terms of the node representations can be expressed as

$$\begin{aligned}
\mathbf{H}^q &= \text{GCN}(\mathbf{A}, \mathbf{X} + \mathbf{Q}) \\
&= \tilde{\mathbf{A}} \cdot (\mathbf{X} + \mathbf{Q}) \cdot \mathbf{W} \\
&= \tilde{\mathbf{A}} \cdot \mathbf{X} \cdot \mathbf{W} + \tilde{\mathbf{A}} \cdot \mathbf{Q} \cdot \mathbf{W} \\
&= \mathbf{H} + \triangle\mathbf{H}^q.
\end{aligned} \tag{27}$$

Similarly, two terms of the graph representations can be described as

$$\begin{aligned}
\mathbf{h}^q &= \text{sum}(\text{GIN}(\mathbf{A}, \mathbf{X} + \mathbf{Q})) \\
&= \text{sum}((\mathbf{A} + (1 + \epsilon)\mathbf{I}) \cdot (\mathbf{X} + \mathbf{Q}) \cdot \mathbf{W}) \\
&= \text{sum}((\mathbf{A} + (1 + \epsilon)\mathbf{I}) \cdot \mathbf{X} \cdot \mathbf{W}) + \text{sum}((\mathbf{A} + (1 + \epsilon)\mathbf{I}) \cdot \mathbf{Q} \cdot \mathbf{W}) \\
&= \mathbf{h} + \triangle\mathbf{h}^q.
\end{aligned} \tag{28}$$

Next, to establish Theorem 3.1, Lemma D.1 is introduced.

**Lemma D.1.** *For any GA $t(\cdot)$ applied to the input graph $\mathcal{G}(\mathbf{A}, \mathbf{X})$, it can be decoupled to a series of three augmentations: attribute augmentation ($t_a(\mathbf{X})$), edge augmentation ($t_e(\mathbf{A})$), and subgraph augmentation ($t_s(\mathbf{A}, \mathbf{X})$) containing node augmentation [8].*

Accordingly, the candidate spaces (*i.e.*, $\mathbb{A}$ and $\mathbb{X}$) can be created through these three augmentations. Theorem 3.1 can be proven based on this lemma by establishing the following three propositions.

**Proposition D.2.** *From the message-passing perspective of the graph encoder $g_{\Theta}(,)$, the proposed implementation of UGA, which is expressed as Eq. 4, can be equivalent to any attribute augmentation $t_a(\mathbf{X})$, that is*

$$g_{\Theta}(\mathbf{A}, \mathbf{X} + \mathbf{Q}) = g_{\Theta}(\mathbf{A}, \mathbf{X}^{(*)}), \tag{29}$$

*where $\mathbf{X}^{(*)} = t_a(\mathbf{X})$ stands for the augmented node attirbutes.*

*Proof.* Firstly, let us discuss the equivalence for node representations. To establish Eq. 29, the goal is to identify $\mathbf{Q}$ such that $\mathbf{H}^q = \mathbf{H}^a$. Let $\triangle\mathbf{X} = \mathbf{X} - \mathbf{X}^{(*)}$ denote the variance in attributes resulting from attribute augmentation, $\mathbf{H}^a$ can be decomposed into two terms: the original representations $\mathbf{H}$ and the augmented representations $\triangle\mathbf{H}^a$. This decomposition can be expressed as

$$\begin{aligned}
\mathbf{H}^a &= \text{GCN}(\mathbf{A}, \mathbf{X} + \triangle\mathbf{X}) \\
&= \tilde{\mathbf{A}} \cdot (\mathbf{X} + \triangle\mathbf{X}) \cdot \mathbf{W} \\
&= \tilde{\mathbf{A}} \cdot \mathbf{X} \cdot \mathbf{W} + \tilde{\mathbf{A}} \cdot \triangle\mathbf{X} \cdot \mathbf{W} \\
&= \mathbf{H} + \triangle\mathbf{H}^a.
\end{aligned} \tag{30}$$

Therefore, the proof shifts from establishing $\mathbf{H}^q = \mathbf{H}^a$ to demonstrating $\Delta\mathbf{H}^q = \Delta\mathbf{H}^a$. It becomes evident that this equivalence holds true under the condition $q_{i,j} = \Delta x_{i,j}$, thus ensuring that $\tilde{\mathbf{A}}\mathbf{Q}\mathbf{W} = \tilde{\mathbf{A}}\triangle\mathbf{X}\mathbf{W}$.

Moreover, this derivation highlights that the conclusion remains consistent regardless of the encoder chosen. Hence, the solution $q_{i,j} = \triangle x_{i,j}$ holds for the node representations encoded using GIN.

Besides, the conclusions drawn at the node level, unaffected by the choice of readout functions (*e.g.*, mean and sum), are equally applicable to the graph level. This insight extends our solution to graph representations $\mathbf{h}$, thereby completing the proof.

$\square$

**Proposition D.3.** *From the message-passing perspective of the graph encoder $g_\Theta(,)$, the proposed implementation of UGA, which is expressed as Eq. 4, can be equivalent to any edge augmentation $t_e(\mathbf{A})$, that is*

$$g_\Theta\left(\mathbf{A}, \mathbf{X} + \mathbf{Q}\right) = g_\Theta\left(\mathbf{A}^{(*)}, \mathbf{X}\right), \tag{31}$$

*where $\mathbf{A}^{(*)} = t_e(\mathbf{A})$ denotes the adjacency matrix obtained from edge augmentation.*

*Proof.* Consistent with the method adopted in the above proofs, the edge-augmented representations ($\mathbf{H}^e$ and $\mathbf{h}^e$) are first computed. Next, the equivalence of these representations with UGA counterparts ($\mathbf{H}^q$ and $\mathbf{h}^q$), respectively, is demonstrated.

Let $\triangle\mathbf{A} = \mathbf{A} - \mathbf{A}^{(*)}$ denotes the topology variance caused by edge augmentation, $\mathbf{H}^e$ can be decoupled into two terms as follows.

$$\begin{aligned}
\mathbf{H}^e &= \text{GCN}\left(\mathbf{A} + \triangle\mathbf{A}, \mathbf{X}\right) \\
&= \left(\tilde{\mathbf{A}} + \triangle\tilde{\mathbf{A}}\right) \cdot \mathbf{X} \cdot \mathbf{W} \\
&= \tilde{\mathbf{A}} \cdot \mathbf{X} \cdot \mathbf{W} + \triangle\tilde{\mathbf{A}} \cdot \mathbf{X} \cdot \mathbf{W} \\
&= \mathbf{H} + \triangle\mathbf{H}^e.
\end{aligned} \tag{32}$$

where $\triangle\tilde{\mathbf{A}} = \tilde{\mathbf{A}} - \tilde{\mathbf{A}}^{(*)}$ terms the topology variance between the adjacency matrix and its augmented version, both of which are normalized. Therefore, this demonstration solely necessitates establishing the equivalence between $\triangle\mathbf{H}_q$ and $\triangle\mathbf{H}_e$, particularly

$$\tilde{\mathbf{A}} \cdot \mathbf{Q} \cdot \mathbf{W} = \triangle\tilde{\mathbf{A}} \cdot \mathbf{X} \cdot \mathbf{W}. \tag{33}$$

Through the utilization of Cayley-Hamilton theorem [6], which asserts that every matrix adheres to its own characteristic polynomial, that is

$$\Gamma(\mathbf{A}) = \mathbf{A}^n + c_1\mathbf{A}^{n-1} + c_2\mathbf{A}^{n-2} + \cdots + c_n\mathbf{I} = 0, \tag{34}$$

where $\{c_i\}_i^n = 1$ stands for the set of polynomial coefficients. Thus, the inverse of matrix $\tilde{\mathbf{A}}$ can be expressed as

$$\tilde{\mathbf{A}}^{-1} = -\frac{1}{c_n}\tilde{\mathbf{A}}^{n-1} - \frac{c_1}{c_n}\tilde{\mathbf{A}}^{n-2} - \cdots - \frac{c_{n-1}}{c_n}\mathbf{I}. \tag{35}$$

Based on it, the equivalent between the proposed UGA and edge augmentation (denoted as Eq. 33) can be demonstrated if it holds that

$$q_{i,j} = [(-\frac{1}{c_n}\tilde{\mathbf{A}}^{n-1} - \frac{c_1}{c_n}\tilde{\mathbf{A}}^{n-2} - \cdots - \frac{c_{n-1}}{c_n}\mathbf{I}) \cdot \triangle\tilde{\mathbf{A}} \cdot \mathbf{X}]_{i,j}. \tag{36}$$

Furthermore, the solution derived for node representations is directly applicable to graph representations as well, with the sole modification being the substitution of $\tilde{\mathbf{A}}$ with $\mathbf{A} + (1 + \epsilon\mathbf{I})$. In light of the above analysis, this proposition is proven. $\square$

**Proposition D.4.** *From the message-passing perspective of the graph encoder $g_\Theta(,)$, the proposed implementation of UGA, which is expressed as Eq. 4, can be equivalent to any subgraph augmentation $t_s(\mathbf{A}, \mathbf{X})$, that is*

$$g_\Theta(\mathbf{A}, \mathbf{X} + \mathbf{Q}) = g_\Theta(\mathbf{A}^{(*)}, \mathbf{X}^{(*)}), \tag{37}$$

*where $\mathbf{A}^{(*)}, \mathbf{X}^{(*)} = t_s(\mathbf{A}, \mathbf{X})$ stand for the adjacency matrix and attribute matrix obtained from subgraph augmentation.*

*Proof.* Note that the subgraph augmentation is typically tailored for graph-level tasks. For node-level tasks, specific subgraph augmentation (where the number of nodes remains unchanged) can be regarded as a type of edge or attribute augmentation, such as masking all attributes of nodes in the

perturbated subgraph. Hence, based on Proposition D.2 and Proposition D.3, it is not hard to conclude that this proposition holds.

Recall that within our UGA, the graph representation $\mathbf{h}_q \in \mathbb{R}^{1 \times d}$ is articulated as $\mathbf{h}_q = \mathbf{h} + \triangle \mathbf{h}_q$. This can be formulated as

$$
\begin{aligned}
\mathbf{h}^q &= \mathrm{sum}\left(\mathrm{GIN}\left(\mathbf{A}, \mathbf{X} + \mathbf{Q}\right)\right) \\
&= \mathrm{sum}\left((\mathbf{A} + (1 + \epsilon)\mathbf{I}) \cdot \mathbf{X} \cdot \mathbf{W}\right) + \mathrm{sum}\left((\mathbf{A} + (1 + \epsilon)\mathbf{I}) \cdot \mathbf{Q} \cdot \mathbf{W}\right) \\
&= \mathbf{h} + \triangle \mathbf{h}^q.
\end{aligned}
\tag{38}
$$

Furthermore, let us assume that the original graph encompasses $k$ subgraphs. The corresponding subgraph-augmented graph, in turn, encompasses $m - 1$ subgraphs, which can be formulated as

$$
\mathbf{A} = \begin{bmatrix} \mathbf{A}_0 & 0 & \cdots & 0 \\ 0 & \mathbf{A}_1 & \cdots & 0 \\ \vdots & \vdots & \ddots & \vdots \\ 0 & 0 & \cdots & \mathbf{A}_{m-1} \end{bmatrix}, \quad \mathbf{X} = \begin{bmatrix} \mathbf{X}_0 \\ \mathbf{X}_1 \\ \vdots \\ \mathbf{X}_{m-1} \end{bmatrix},
\tag{39}
$$

where $\mathbf{A}_i$ denotes the adjacency matrix of the $i$-th subgraph and $\mathbf{X}_i$ terms the corresponding attribute matrix. Thus, for the augmented subgraphs $\{\mathcal{G}(\mathbf{A}_t, \mathbf{X}_t)\}_{i=0}^{m-1}$, the graph representation $\mathbf{h}^s$ can be formulated as

$$
\mathbf{h}^s = \sum_{t=0}^{m-1} \mathrm{sum}((\mathbf{A}_t + (1 + \epsilon)\mathbf{I}) \cdot \mathbf{X}_t \cdot \mathbf{W}),
\tag{40}
$$

Let $\triangle \mathbf{h} = \mathbf{h}^s - \mathbf{h}$ denotes the representation variance caused by subgraph augmentation, there is

$$
\triangle \mathbf{h} = \sum_{t=0}^{m-1} \Omega_t \cdot \mathrm{sum}((\mathbf{A}_t + (1 + \epsilon)\mathbf{I}) \cdot \mathbf{X}_t \cdot \mathbf{W}),
\tag{41}
$$

where $\Omega$ denotes an indicator vector. If $m - 1 < k$, the subgraph augmentation typically refers to the subgraph corrupting (denoted as Eq. 20). Thus, for the $t$-th perturbated subgraph, there is $\Omega_t = -1$. And if $m - 1 > k$, is generally understood as the subgraph inserting (formulated as Eq. 21), $\Omega_t = 1$. Additionally, if the subgraph is not changed, $\Omega_t = 0$.

Next, we aim to identify a solution for $\mathbf{q}$ that satisfies the following conditions:

$$
\triangle \mathbf{h}^q = \triangle \mathbf{h},
\tag{42}
$$

This can be further formulated as

$$
\begin{aligned}
\mathrm{sum}\left((\mathbf{A} + (1 + \epsilon)\mathbf{I}) \cdot \mathbf{Q} \cdot \mathbf{W}\right) &= \sum_{t=0}^{m-1} \Omega_t \cdot \mathrm{sum}\left((\mathbf{A}_t + (1 + \epsilon)\mathbf{I}) \cdot \mathbf{X}_t \cdot \mathbf{W}\right) \\
&= \mathbf{d}^\top \cdot \mathbf{Q} \cdot \mathbf{W} = \sum_{t=0}^{m-1} \Omega_t \cdot \mathrm{sum}\left((\mathbf{A}_t + (1 + \epsilon)\mathbf{I}) \cdot \mathbf{X}_t\right) \cdot \mathbf{W},
\end{aligned}
\tag{43}
$$

where $\mathbf{d} = [d_0 + 1 + \epsilon, \ldots, d_{n-1} + 1 + \epsilon] \in \mathbb{R}^n$ stands for a degree vector.

In the general case, assuming the absence of isolated nodes within the graph, the degree vector $\mathbf{d}$ of the input graph is devoid of zero elements. Consequently, a solution for $\mathbf{Q}$ can be formulated as

$$
\mathbf{Q} = \mathbf{D}^+ \tilde{\mathbf{H}},
\tag{44}
$$

where $\tilde{\mathbf{H}} = \sum_{t=0}^{m-1} \Omega_t \cdot \mathrm{sum}((\mathbf{A}_t + (1 + \epsilon)\mathbf{I}) \cdot \mathbf{X}_t)$ and $\mathbf{D}^+ = \frac{1}{\mathbf{d}\mathbf{d}^\top} \mathbf{d}$ denotes the Moore-Penrose pseudoinverse of $\mathbf{d}$. Thus, equivalence can be established if each element $q_{i,j}$ in AC matrix $\mathbf{Q}$ satisfies the following condition:

$$
q_{i,j} = [\mathbf{D}^+ \cdot \sum_{t=0}^{m-1} \mathrm{sum}\left((\mathbf{A}_t + (1 + \epsilon\mathbf{I})) \cdot \mathbf{X}_t\right)]_{i,j}.
\tag{45}
$$

Therefore, the proof ends. $\qquad\square$

**Extension to multi-layer graph encoders.** Following the discussion on single-layer graph encoders, solutions for multi-layer graph encoders are identified. Initially, as discussed in [18], various GNNs can be formulated as

$$\mathbf{H} = \mathbf{S} \cdot \mathbf{X} \cdot \mathbf{W}, \tag{46}$$

where $\mathbf{S}$ denotes the diffusion matrix, exemplified by $\tilde{\mathbf{A}}$ for GCN and $\mathbf{A} + (1 + \epsilon)\mathbf{I}$ for GIN. In addition, $\mathbf{W}$ represents the projection layer, such as the linear projection utilized in both GCN and GIN. Here, the nonlinear activation function between layers is not considered. With this architecture, the node representations at the $k$-th layer can be formulated as

$$\mathbf{H} = g_\Theta^l(\mathbf{S}, \mathbf{X}) = \mathcal{S} \cdot \mathbf{X} \cdot \mathcal{W}, \tag{47}$$

where $\mathcal{S} = \prod_{i=0}^l \mathbf{S}^l$ terms the product of adjacency matrices. Similarly, $\mathcal{W} = \prod_{i=0}^l \mathbf{W}^l$ represents the product of linear projection matrices. Given that the aforementioned conclusions are independent of the forms of these two matrices, it is not hard to prove that by substituing $\mathbf{A}$ and $\mathbf{W}$ with $\mathcal{S}$ and $\mathcal{W}$ in Eq. 25 and Eq. 26, respectively, Proposition D.2, D.3, and D.4 still hold true.

## D.2 Proofs for Theorem 3.4

*Proof.* Firstly, note that two augmented graphs are from the same input graph $\mathcal{G}(\mathbf{A}, \mathbf{X})$ without loss of the information in this graph (especially the edges and attributes). Therefore, there is $\mathbf{A}^{(1)} = \mathbf{A}^{(2)}$ and $\mathbf{X}^{(1)} = \mathbf{X}^{(2)}$. Accordingly, the diversity can be transformed into

$$\mathcal{D}_v = \| \sum_{t \in \mathcal{N}_v \cup v} (\mathbf{x}_{t,:}^{(1)} + \hat{\mathbf{q}}_{t,:}) - \sum_{t \in \mathcal{N}_v \cup v} (\mathbf{x}_{t,:}^{(2)} + \hat{\mathbf{p}}_{t,:}) \|_F^2 \tag{48}$$

$$= \| \sum_{t \in \mathcal{N}_v \cup v} \hat{\mathbf{q}}_{t,:} - \sum_{t \in \mathcal{N}_v \cup v} \hat{\mathbf{p}}_{t,:} \|_F^2 \tag{49}$$

$$= \| \sum_{t \in \mathcal{N}_v \cup v} \mathbf{b}_{t,:}^q \mathbf{Q} - \sum_{t \in \mathcal{N}_v \cup v} \mathbf{b}_{t,:}^p \mathbf{P} \|_F^2. \tag{50}$$

Given the conditions $\mathbf{b}_{t,:}^q = \sigma(\mathbf{x}_{t,:}^{(1)} \mathbf{Q}^\top)$ where $\sigma$ denotes the softmax function, we can calculate that

$$\mathcal{D}_v = \| \sum_{t \in \mathcal{N}_v \cup v} \mathbf{b}_{t,:}^q \mathbf{Q} - \sum_{t \in \mathcal{N}_v \cup v} \mathbf{b}_{t,:}^p \mathbf{P} \|_F^2 \tag{51}$$

$$= \| \sum_{t \in \mathcal{N}_v \cup v} \sigma(\mathbf{x}_{t,:}^{(1)} \mathbf{Q}^\top) \mathbf{Q} - \sum_{t \in \mathcal{N}_v \cup v} \sigma(\mathbf{x}_{t,:}^{(2)} \mathbf{P}^\top) \mathbf{P} \|_F^2 \tag{52}$$

$$= \| (\sum_{t \in \mathcal{N}_v \cup v} \sigma(\mathbf{x}_{t,:}^{(1)} \mathbf{Q}^\top)) \mathbf{Q} - (\sum_{t \in \mathcal{N}_v \cup v} \sigma(\mathbf{x}_{t,:}^{(2)} \mathbf{P}^\top)) \mathbf{P} \|_F^2. \tag{53}$$

For clarity, $\mathbf{x}_{t,:}$ is employed to represent both $\mathbf{x}_{t,:}^{(1)}$ and $\mathbf{x}_{t,:}^{(2)}$. In light of the consistency constraint within the GOUDA framework, the difference between $\mathbf{Q}$ and $\mathbf{P}$ is minimal, which can be interpreted as a minor perturbation $\Delta$, such that $\mathbf{Q} = \mathbf{P} + \Delta$. Therefore, Eq. 53 can be further reformulated as

$$\mathcal{D}_v = \| (\sum_{t \in \mathcal{N}_v \cup v} \mathbf{x}_{t,:} \mathbf{Q}^\top) \mathbf{Q} - (\sum_{t \in \mathcal{N}_v \cup v} \mathbf{x}_{t,:} \mathbf{P}^\top) \mathbf{P} \|_F^2 \tag{54}$$

$$= \| (\sum_{t \in \mathcal{N}_v \cup v} \mathbf{x}_{t,:} (\mathbf{P} + \Delta^\top)(\mathbf{P} + \Delta) - (\sum_{t \in \mathcal{N}_v \cup v} \mathbf{x}_{t,:} \mathbf{P}^\top) \mathbf{P} \|_F^2. \tag{55}$$

Given that $\Delta$ is considered to be small, the terms $\Delta^2$ and $\mathbf{x}_{t,:} \Delta^\top$ can be neglected. Moreover, it can be assumed that the product of $\mathbf{x}_{t,:} \Delta^\top$ with $\mathbf{P}$ and $\Delta$ is insignificant in comparison to the other terms. Hence, the above formulation can be simplified as

$$\mathcal{D}_v \approx \| (\sum_{t \in \mathcal{N}_v \cup v} \mathbf{x}_{t,:} \mathbf{P}^\top) \Delta \|_F^2. \tag{56}$$

Since $(\sum_{t \in \mathcal{N}_v \cup v} \mathbf{x}_{t,:} \mathbf{P}^\top)$ is a constant matrix, it can be factored out, resulting in:

$$\mathcal{D}_v \approx (\sum_{t \in \mathcal{N}_v \cup v} \mathbf{x}_{t,:} \mathbf{P}^\top)^2 \|\Delta\|_F^2$$
$$= (\sum_{t \in \mathcal{N}_v \cup v} \mathbf{x}_{t,:} \mathbf{P}^\top)^2 \|\mathbf{Q} - \mathbf{P}\|_F^2. \tag{57}$$

Hence, taking into account that $\left(\sum_{t \in \mathcal{N}_v \cup v} \mathbf{x}_{t,:} \mathbf{P}^\top\right)^2$ serves as the proportionality constant, which depends on the values of $\mathbf{x}_{t,:}$ and $\mathbf{P}$, we can deduce that

$$\mathcal{D}_v \approx \|\mathbf{Q} - \mathbf{P}\|_F^2. \tag{58}$$

Based on the above analysis, we conclude the proof. $\square$

## E    Experimental Details

### E.1    Introduction of Datasets

**Datasets for node-level tasks.**

- Citation networks [28]: Cora, Citeseer, PubMed. Each node in these networks represents a scholarly article, with edges indicating citation relationships. Nodes are defined by attributes such as abstracts, keywords, full-text content, and derived features like TF-IDF vectors. Labels to which nodes belong, typically corresponding to research areas or topics.
- Reference network [25]: Wiki-CS. This network represents a collection of Wikipedia articles in computer science. Each node corresponds to an article, characterized by its text and hyperlinks, while the edges depict hyperlinked references between articles. Node labels denote specific subfields of computer science covered by the articles.
- Co-purchase networks [29]: Amazon Photo (Photo for short), Amazon Computers (Computers for short). Nodes represent products available for purchase, with attributes such as features, prices, and customer reviews. Node labels correspond to product types or brands, while edges indicate co-purchase relationships, reflecting the frequency of items commonly bought together by customers.
- Co-author network [29]: Coauthor Physics (Physics for short). Nodes represent physicists, each described by their publication record, research interests, and affiliations. Labels indicate distinct areas or subfields within physics. Edges between nodes stand for collaborations, typically formed through joint publications or co-authorship of scientific papers.

**Datasets for graph-level tasks.**

- Collaborative movie networks [42]: IMDB-BINARY (IMDB-B for short) and IMDB-MULTI (IMDB-M for short). Nodes denote actors or actresses, and an edge exists between two nodes if the individuals have co-starred in the same film.
- Scholarly collaboration network [42]: COLLAB. Researchers as nodes and edges indicate partnerships between them.

It is important to mention that node attributes are absent in the three datasets for graph-level tasks, making one-hot encoding of the degree a typical approach.

We source these datasets from the public repository PyTorch Geometric (PyG) [9]. The datasets can be accessed through the URLs listed below:

- Cora, CiteSeer, PubMed: https://github.com/kimiyoung/planetoid/raw/master/data.
- Wiki-CS: https://github.com/pmernyei/wiki-cs-dataset/raw/master/dataset.
- Photo, Computers: https://github.com/shchur/gnn-benchmark/raw/master/data/npz/.
- Physics: https://github.com/shchur/gnn-benchmark/raw/master/data/npz/.
- IMDB-B, IMDB-M, COLLAB: https://ls11-www.cs.tu-dortmund.de/staff/morris/graph kerneldatasets.

### E.2    Dataset Splitting

For the seven benchmark datasets utilized for node classification tasks (Cora, Citeseer, PubMed, Wiki-CS, Computers, Photo, and Physics), the dataset is divided into training, validation, and testing sets in the ratio of 1:1:8. For the three benchmark datasets employed for graph classification tasks, namely IMDB-B, IMDB-M, and COLLAB, a 10-fold cross-validation approach is adopted to partition.

Table 5: Statistics of ten graph benchmark datasets.

| | Node-level tasks | | | | | | | Graph-level tasks | | |
|---|---|---|---|---|---|---|---|---|---|---|
| | Cora | CiteSeer | PubMed | Wiki-CS | Computers | Photo | Physics | IMDB-B | IMDB-M | COLLAB |
| # Graphs | 1 | 1 | 1 | 1 | 1 | 1 | 1 | 1,000 | 1,500 | 5,000 |
| # Nodes | 2,708 | 3,327 | 19,717 | 11,701 | 13,752 | 7,650 | 34,493 | 19.8 | 13.0 | 74.5 |
| # Edges | 5,429 | 4,732 | 44,338 | 216,123 | 245,861 | 119,081 | 991,848 | 193.1 | 66.0 | 4914.4 |
| # Features | 1,433 | 3,703 | 500 | 300 | 767 | 745 | 8,451 | - | - | - |
| # Classes | 7 | 6 | 3 | 10 | 10 | 8 | 5 | 2 | 3 | 3 |

### E.3 Introduction of Baselines

**Baselines for node-level tasks.** Details on the baselines for node-level tasks are outlined below.

GCN [17]: It is a representative Graph Neural Network (GNN) that utilizes spectral and spatial strategies to perform graph convolutional operations. It makes each node aggregate information from its neighbor nodes by integrating the graph topology and node attributes.

GAT [34]: It introduces an attention mechanism into the GNN, enabling each node to weigh the importance of its neighbor nodes during the aggregation process.

Unsupervised baselines are detailed below.

DGI [35]: An Infomax principle-based GCL augments the graph via row-wise shuffling of the attribute matrix and maximizes the mutual information between global and local representations.

GMI [26]: It is a variant model of DGI that maximizes a comprehensive graphical mutual information metric, including features and edges between nodes in both the input and reconstructed output graphs.

MVGRL [14]: It is a variant model of DGI, employing contrastive learning between various structural views of graphs, including first-order adjacency and graph diffusions.

GRACE [53]: It is a GCL model that generates node embeddings by corrupting both graph structure (via random edge removing) and attributes (via random attribute masking) to create diverse views and maximize their agreement.

GCA [54]: It is a variant model of GRACE, which incorporates adaptive augmentation strategies based on node centrality, aiming to enhance the flexibility of the model.

BGRL [32]: It is a GCL model that augments the graph through random edge removing and employs bootstrapping to update the parameters of the online encoder.

GBT [2]: A feature-level GCL model leverages the validated Barlow Twins loss for training, aiming to reduce redundant information between two views obtained through random edge removing.

CCA-SSG [50]: It is a GCL model utilizing feature-level contrast derived from Canonical Correlation Analysis (CCA) to learn the node representations.

SPAN [21]: It introduces a spectral augmentation scheme for topology augmentation by perturbing the graph spectrum, and aiming to maintain spectral invariance to sensitive structures, and minimizing graph spectrum changes.

DSSL [39]: It is a graph self-supervised model that presents an approach that disentangles the varied neighborhood contexts of a node, aiming to model multifaceted information within the graph.

HomoGCL [20]: It is a localized variant model of GRACE, incorporating k-mean-based saliency values to weigh the importance of neighbor nodes.

**Baselines for graph-level tasks.** Introductions of the baselines for graph-level tasks are given below.

InfoGraph [30]: It is a variant of DGI, which maximizes mutual information between graph-level representations and substructures at different scales, such as nodes, edges, and triangles.

GraphCL [48]: It is a GCL framework that learns graph representations by employing various augmentation techniques on the local subgraphs of nodes.

JOAO [47]: It is a variant of GraphCL, which utilizes the min-max optimization to automatically select the most effective GAs during the contrastive learning process.

AD-GCL [31]: It is a GCL framework based on adversarial training, introducing an attack process to modify the edges.

For the baseline implementations, we utilize PyG to implement GCN and GAT. In addition, for the self-supervised baselines, we employ their source codes. The sources are listed below:

- GCN, GAT: https://github.com/pyg-team/pytorch_geometric/tree/master/torch_geometric/nn/conv.
- DGI: https://github.com/PetarV-/DGI.
- GMI: https://github.com/zpeng27/GMI.
- MVGRL: https://github.com/kavehhassani/mvgrl.
- GRACE: https://github.com/CRIPAC-DIG/GRACE.
- GCA: https://github.com/CRIPAC-DIG/GCA/.
- BGRL: https://github.com/nerdslab/bgrl.
- GBT: https://github.com/pbielak/graph-barlow-twins.
- CCA-SSG: https://github.com/hengruizhang98/CCA-SSG.
- SPAN: https://github.com/haonan3/spgcl.
- DSSL: https://papers.nips.cc/paper_files/paper/2022/file/040c816286b3844fd78f2124eec75f2e-Supplemental-Conference.zip.
- HomoGCL: https://github.com/wenzhilics/HomoGCL
- InfoGraph: https://github.com/sunfanyunn/InfoGraph
- GraphCL: https://github.com/Shen-Lab/GraphCL.
- JOAO: https://github.com/Shen-Lab/GraphCL_Automated.
- AD-GCL: https://github.com/susheels/adgcl.

### E.4 Complexity Analysis

This subsection analyzes the complexity of GOUDA in comparison with the baseline GCL equipped with learnable GAs (*i.e.*, SPAN, JOAO, and AD-GCL). Note that since the updates of these GCLs can utilize the same graph encoder (*e.g.*, GCN) and contrastive loss (*e.g.*, InfoNCE loss), this discussion focuses solely on the time complexity of the augmentation phase.

To enhance clarity, let us define our terms: $n$ represents the number of nodes corresponding to the network size; $m$ signifies the number of edges; $f$ refers to the dimension of attributes; and $d$ denotes the dimension of the hidden layers.

**SPAN** [21]: The time complexity for augmentations in SPAN is $O(n^2 tk)$, where $t$ denotes the time of iterations and $k$ is the number of eigenvalues to be selected. Specifically, SPAN necessitates an iterative optimization of the augmentation through eigen-decomposition, which demands a $O(tn^3)$ complexity. Nonetheless, this complexity can be reduced to $O(n^2 tk)$ by employing selective eigen-decomposition on the $k$ lowest- and highest-eigenvalues via the Lanczos Algorithm.

**JOAO** [47]: The time complexity for augmentations in JOAO is $O(n^2 d)$. Specifically, JOAO employs a min-max optimization strategy to refine the parameters that are used for selecting augmentations from an option pool. This process involves maximizing the contrastive loss, which inherently requires a $O(n^2 d)$ complexity.

**AD-GCL** [31]: The time complexity for augmentations in AD-GCL is $O(n^2 d)$. Similarly, AD-GCL employs a min-max optimization strategy, but its objective is to modify edges. Therefore, this process also entails maximizing the contrastive loss, which carries a $O(n^2 d)$ complexity. In addition, encoding edges is required, which introduces a complexity of $O(md^2)$.

**GOUDA** : The time complexity for augmentations in the proposed GOUDA is $O(nkf)$. GOUDA introduces $k$ AC vectors to nodes and utilizes the consistency-diversity balance principle to update these AC vectors, where $k \ll n$. Firstly, nodes are updated by aggregating the features of AC vectors, which incurs a complexity of $O(nkf)$. Besides, GOUDA maintains consistency by minimizing the contrastive loss, a process inherent to the training model and thus does not introduce additional

complexity; to ensure diversity, it calculates the independence loss, which brings in a complexity of $O(k^2 f)$. Therefore, the overall complexity is $O(nkf)$.

In summary, GOUDA presents a more computationally efficient approach than the baselines, which is supported by the evidence presented in Section 4.

### E.5 Configurations and Hyper-parameters

#### E.5.1 Configurations

The experiments leverage the linear evaluation method [26], where models are firstly trained in an unsupervised manner, and then, the obtained embeddings are utilized for downstream tasks. For the node-level tasks, the graph encoder $g_\Theta$ is configured as a two-layer GCN [17], while for the graph classification tasks, it is set as a five-layer GIN [40]. In the evaluation phase, we utilize a single-layer linear classifier [27] for node classification [26], apply K-means [16] to node embeddings for node clustering, and train an SVM classifier [3] for graph classification. The results for node-level tasks are the average of ten random runs, while those for graph-level tasks are based on five runs.

#### E.5.2 Environment

All experiments are conducted on two Linux machines as shown in Tab. 6.

Table 6: Experimental environment servers.

|  | Server 1 | Server 2 |
|---|---|---|
| OS | Linux 5.15.0-82-generic | Linux 5.15.0-100-generic |
| CPU | Intel(R) Core(TM) i7-12700K CPU @ 3.6GHz | Intel(R) Core(TM) i9-10980XE CPU @ 3.00GHz |
| GPU | GeForce RTX 4090 | GeForce RTX 3090 |

#### E.5.3 Hyper-parameter Settings

GOUDA is implemented as two models: **GOUDA-IF**, which utilizes InfoNCE loss, and **GOUDA-BT**, which employs BarlowTwins loss. For the node-level tasks, both models are trained using an Adam optimizer with a learning rate of $1e^{-3}$ and the weight decay rate from $\{0, 5e^{-5}, 5e^{-4}\}$. The dimensions $d$ of node embeddings are selected from $\{256, 512, 1024, 2048\}$, and their impact is analyzed in Section 4.2. The hyperparameters $\beta_1$ and $\beta_2$ of independence loss are chosen from $\{1e^{-4}, 1e^{-3}\}$, while the related hyperparameter $\gamma$ is selected among $\{1e^{-2}, 1e^{-1}, 1, 10\}$. Additionally, for GOUDA-IF, the temperature coefficient $\tau$ is selected from $\{0.2, 0.4, 0.6, 0.8\}$, while for GOUDA-BT, the hyperparameter $\lambda$ is set to $\frac{1}{d}$. For the graph-level task, the configuration follows GraphCL [48], where the hidden dimension is fixed to 128 and the penalty parameter of SVM is selected from $\{1e^{-3}, 1e^{-2}, 1e^{-1}, 1, 1e^2, 1e^3\}$. The choice of threshold $\epsilon$ is given in Section E.6.2.

### E.6 Additional Experiment Results

#### E.6.1 Complete Results for Node Clustering

Tab. 7 presents the comprehensive results of the node clustering experiments. The analysis of these results can be found in Section 4.

#### E.6.2 Other Hyperparameter Analysis

**Embedding Dimension.** This experiment aims to shed light on the selection of hyperparameter $d$. As depicted in Fig. 8, the proposed GOUDA shows improved performance with an increased embedding dimension. Notably, GOUDA-IF and GOUDA-BT exhibit reduced performance at an embedding dimension of 256 compared to higher dimensions. This indicates that a large embedding dimension is essential for contrastive learning models to capture robust representations. Moreover, it is observable that there is a slight decrease in the performance of GOUDA-IF with large dimensions. This can be due to overfitting to the self-supervised signal, which may hinder its generalization capability.

Table 7: Node clustering performance: NMI & ARI Scores in percentage (mean$_{\pm\text{std}}$).

| | Cora | | CiteSeer | | PubMed | |
| | NMI | ARI | NMI | ARI | NMI | ARI |
|---|---|---|---|---|---|---|
| DGI | $52.75_{\pm0.94}$ | $47.78_{\pm0.65}$ | $40.43_{\pm0.81}$ | $41.84_{\pm0.62}$ | $30.03_{\pm0.50}$ | $29.78_{\pm0.28}$ |
| MVGRL | $54.21_{\pm0.25}$ | $49.04_{\pm0.67}$ | $43.26_{\pm0.48}$ | $42.73_{\pm0.93}$ | $30.75_{\pm0.54}$ | $30.42_{\pm0.45}$ |
| GRACE | $54.59_{\pm0.32}$ | $48.31_{\pm0.63}$ | $43.02_{\pm0.43}$ | $42.32_{\pm0.81}$ | $31.11_{\pm0.48}$ | $30.37_{\pm0.51}$ |
| GBT | $55.32_{\pm0.65}$ | $48.91_{\pm0.73}$ | $44.01_{\pm0.97}$ | $42.61_{\pm0.63}$ | $31.33_{\pm0.57}$ | $30.64_{\pm0.74}$ |
| CCA-SSG | $56.38_{\pm0.62}$ | $50.62_{\pm0.90}$ | $43.98_{\pm0.94}$ | $42.79_{\pm0.77}$ | $32.06_{\pm0.40}$ | $31.15_{\pm0.85}$ |
| GOUDA-IF | $\mathbf{57.92}_{\pm0.49}$ | $\mathbf{52.41}_{\pm0.58}$ | $\mathbf{45.11}_{\pm0.79}$ | $\mathbf{43.82}_{\pm0.65}$ | $\mathbf{33.17}_{\pm0.45}$ | $\mathbf{31.98}_{\pm0.46}$ |
| GOUDA-BT | $\underline{57.35}_{\pm0.51}$ | $\underline{51.84}_{\pm0.61}$ | $\underline{44.93}_{\pm0.85}$ | $\underline{43.46}_{\pm0.71}$ | $\underline{33.14}_{\pm0.51}$ | $\underline{31.73}_{\pm0.52}$ |

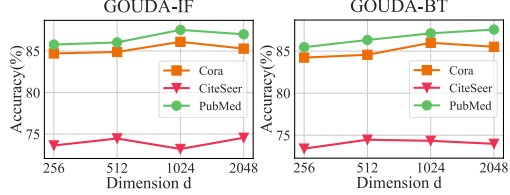

Figure 8: Impact of dimension $d$.

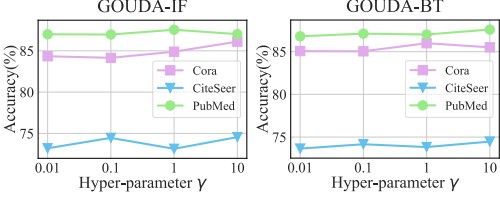

Figure 9: Impact of hyperparameter $\gamma$.

**Weight of Independence Loss.** Several insights are yielded from observations in Fig. 9. Firstly, the proposed GOUDA shows stability regardless of parameter $\gamma$ changes. Secondly, the framework maintains robust performance even at a low value of 0.01 and 0.1. However, the omission of the proposed Independence loss does have a detrimental effect, as demonstrated by the Ablation Study (in Section 4.2). Lastly, while the tested parameter range is $\{1, 10\}$, future research should consider broader or more detailed ranges.

**Threshold for sparsification.** The performance changes resulting from varying the hyperparameter $\epsilon$ are detailed in Tab. 8. To eliminate bias due to network size, $\epsilon$ is not freely tuned. Instead, it is set as the output of the selection function $\epsilon = \text{selection}(\mathbf{B}, s)$, which estimates this threshold. $\mathbf{B}$ denotes the matrix of propagation weights from AC vectors to nodes, and $s$ stands for the proportion of the largest elements retained. The value of $s$ is chosen from the set $\{0.2, 0.4, 0.6, 0.8\}$ in the experiments. From Tab. 8, it can be observed that the threshold does not significantly affect the model performance. Specifically, within the range of selection, the variation in model performance does not exceed $2\%$.

Table 8: Impact of threshold $\epsilon$.

| | GOUDA-IF | | | | GOUDA-BT | | | |
| | 0.2 | 0.4 | 0.6 | 0.8 | 0.2 | 0.4 | 0.6 | 0.8 |
|---|---|---|---|---|---|---|---|---|
| Cora | 85.29 | 84.71 | 84.19 | **86.11** | 85.07 | 84.56 | 85.88 | **85.99** |
| CiteSeer | **74.55** | 73.20 | 73.11 | 73.62 | 74.34 | **74.47** | 73.98 | 73.41 |
| PubMed | 86.96 | 87.25 | **87.55** | 87.05 | **87.59** | 86.94 | 86.76 | 87.11 |

