# OpenReview forum: "Unified Graph Augmentations for Generalized Contrastive Learning on Graphs"
_NeurIPS.cc/2024/Conference — NeurIPS 2024 poster_

### Official Review · Reviewer_sGNH · 2024-07-01

**Soundness:** 3
**Presentation:** 3
**Contribution:** 3
**Rating:** 8
**Confidence:** 5

**Summary:**

This paper investigates the generality of graph augmentations across various types of graphs and tasks. Firstly, the paper presents a new interpretation that unifies the various graph augmentations into local attribute modifications of each node from a basic message-passing perspective. Then, the paper develops a novel graph augmentation module that alters node attributes to simulate the augmentation effects. Utilizing this module, the paper presents a novel framework for graph contrastive learning, where the balance between consistency and diversity across different augmented views is carefully considered. Finally, it provides a theoretical understanding of the generality of the proposed framework. Extensive evaluations demonstrate its superior performance on diverse datasets and tasks.

**Strengths:**

1) This paper is well-organized and well-written.
2) The motivation of unifying graph augmentations from a message-passing view is interesting.
3) The proposed framework is simple and efficient with solid theoretical guarantee.
4) Experiments across various types of graphs and tasks demonstrate the superior efficacy and efficiency of the proposed method in comparison to existing works.

**Weaknesses:**

1) There are a few typos and grammar mistakes in the paper that need fixing.
2) The theoretical analysis requires intuitive explanations. While the authors offer the proof of generality, they should elucidate how the theorem relates to the superior performance.
3) The concepts of consistency and diversity deserve a more comprehensive explanation. Additionally, how do the definitions of consistency and diversity across augmentations in self-supervised learning scenarios differ from those in semi-supervised learning scenarios?
4) The connection between the number of AC vectors and model performance is not clear.

**Questions:**

Please refer to Weaknesses.

---

> ### Author Rebuttal · Authors · 2024-08-07
>
> ***Q1. There are a few typos and grammar mistakes in the paper that need fixing.***
>
> R1. Thanks for your thoughtful reminder. We will polish our manuscript to elevate the quality and clarity.
>
> ---
>
> ***Q2. While the authors offer the proof of generality, they should elucidate how the theorem relates to the superior performance.***
>
> R2. An intuitive explanation is that the generality improves the flexibility of the augmentation, which is the need for diverse graph tasks. Graphs and corresponding tasks possess diverse characteristics and, hence, need different augmentation strategies to meet their diverse and complicated requirements. However, existing graph augmentations, no matter whether heuristics or learnable ones are limited to selecting from discrete candidate sets, e.g., node or edge dropping and attribute masking; thus, some flexible and continuous augmentations can NOT be achievable to satisfy these diverse needs. Fortunately, the proposed GA module is endowed with theoretical versatility, enabling it to be equivalent to any flexible graph augmentations. Therefore, by equipping with this versatile module, the proposed model can be flexibly tailored to meet the diverse GA demands across different tasks. The versatility and flexibility have propelled our framework to achieve outstanding performance across various datasets and tasks, outperforming existing GCL baselines.
>
> ---
>
> ***Q3. The concepts of consistency and diversity deserve a more comprehensive explanation.***
>
> R3. **Consistency** means that the augmented graph maintains the underlying structure and key features of the original graph. Therefore, GAs should minimally impact the similarity between the representations from different augmented graphs to preserve the intrinsic semantic integrity of samples. In contrast, **diversity** indicates that the augmented graphs originate from diverse distributions. Thus, another objective of GAs is to minimize the overlap between augmented graphs, ensuring the model does not become overly fixated on the specific features of a single distribution.
>
> ---
>
> ***Q4.  How do the definitions of consistency and diversity across augmentations in self-supervised learning scenarios differ from those in semi-supervised learning scenarios?***
>
> R4. The distinction in the concepts of consistency and diversity between self-supervised and semi-supervised learning scenarios primarily stems from the availability of supervision signals.
>
> 1. **Semi-supervised learning**
>
> - Consistency: It is often defined as the extent to which augmented graphs align with the inherent patterns of the labeled data.
>
> - Diversity: It is often defined as the difference in distribution between the augmented and input graphs.
>
> 1. **Self-supervised learning**
>
> - Consistency: It is often defined as the extent to which various augmented graphs preserve the essential characteristics of the input graph.
>
> - Diversity: It is often defined as the distributional differences among the various augmented graphs.
>
> ---
>
> ***Q5. The connection between the number of AC vectors and model performance.***
>
> R5. We recognize your concerns about the selection of the $k$, which is a key hyperparameter in the proposed GOUDA. The proposed framework sets $k$ as a small number, which is independent of the size of the graph. The submitted manuscript has presented an experimental assessment of the impact of varying $k$ on model performance, as shown in Figure 7, where $k$ is selected from the set {1, 5, 10, 20}. Our observations indicate that GOUDA is relatively insensitive to the hyperparameter $k$ as explained in Section 4.2. Therefore, one need not be overly concerned with the precise value of $k$ for GOUDA’s robustness to changes in $k$.

---

> ### Comment · Reviewer_sGNH · 2024-08-10
> **Response to authors**
>
> Thank you for the authors' detailed response. I am satisfied with the authors' response and revised manuscript, so I would like to increase my score from 7 to 8.

---

### Official Review · Reviewer_9GrB · 2024-07-05

**Soundness:** 3
**Presentation:** 3
**Contribution:** 3
**Rating:** 7
**Confidence:** 4

**Summary:**

This paper explores the characteristics of local attribute modification in current graph contrastive learning methods. It then integrates diverse augmentation strategies with attribute learning into a unified framework. This unified approach introduces a novel and straightforward graph contrastive learning framework that establishes consistency between embeddings and ensures diversity through augmentation. Specifically, diversity is enforced using the HSIC term. The proposed framework offers two main advantages: 1) high efficiency and 2) universally learnable augmentation. Its effectiveness is validated across graph-level and node-level tasks, demonstrating its superiority.

**Strengths:**

- The exploration of graph contrastive learning mechanisms from a spatial perspective is insightful.

- The unified augmentation proposed and the subsequent framework are both straightforward and efficient.

- The use of shared AC and HSIC is technically robust.

- The experimental evaluations provide convincing evidence.

**Weaknesses:**

- The symbols $\gets$ used in many equations, such as Eq. (4), should be changed to $\to$ for convenience.

- Some experiments require further explanation. It's unclear why the proposed framework demonstrates robustness to topology and attribute noises.

- Some hyper-parameters remain unverified. It's uncertain whether the parameter $\varepsilon$ in Eq. (7) significantly affects performance.

- As far as I know, the HSIC term is computationally intensive. Does this affect efficiency?

- Descriptions from Lines 143 to 153 are difficult to understand.

- Reference [33] is based on adversarial training rather than adversarial attack.

- Some commas are missing after equations, for instance, in Eq. (9).

**Questions:**

See above.

---

> ### Author Rebuttal · Authors · 2024-08-07
>
> ***Q1. The symbols $\leftarrow$ used in many equations, such as Eq. (4), should be changed to $\rightarrow$ for convenience.***
>
> R1. Following your advice, we will revise the equations to replace the symbols $\leftarrow$ with the symbols $\rightarrow$ to enhance readability.
>
> ---
>
> ***Q2. It's unclear why the proposed framework demonstrates robustness to topology and attribute noises.***
>
> R2. Before we explain the robustness advantages of our proposed framework that employs learnable Graph Augmentation (GA) over the baseline models that utilize static and random GAs, we first describe the impact of noise attacks on graphs. Specifically, noise attacks can potentially destroy the semantic subgraphs crucial for predictions, thereby inducing distortions in the learned representations. However, random GAs struggle to preserve the information-rich subgraphs and may further destroy them, exacerbating semantic bias. In contrast, the proposed learnable GAs, trained with a relevant proxy task like contrastive loss, can preserve such subgraphs and, hence, can mitigate semantic bias.
>
> ---
>
> ***Q3. It's uncertain whether the parameter $\varepsilon$ in Eq. (7) significantly affects performance.***
>
> R3. According to your suggestion, we verify the impact of the hyper-parameter $\varepsilon$ on model performance. $\varepsilon$ stands for the threshold to suppress the elements in $B$ to zeros. Thus, to eliminate bias due to network size,  $\varepsilon$ is not freely tuned. Instead, it is set as the output of the selection function, i.e., $\varepsilon = \text{selection}(B, s)$, which estimates this threshold. $B$ represents the matrix of propagation weights from AC vectors to nodes, and $s$ denotes the presence of the largest elements retained. $s$ is chosen from the set {0.2, 0.4, 0.6, 0.8} in the experiments. Thus, the impact of $s$ on the performance is shown in the following table. It can be observed that $s$ does not significantly affect the model performance.
>
> | GOUDA-IF | 0.2  | 0.4| 0.6 | 0.8 |
> |:--------| :---------:|:--------:|:--------:|:--------:|
> | Cora | 85.29 | 84.71 | 84.19 | 86.11 |
> | CiteSeer | 74.55 | 73.20 | 73.11 | 73.62 |
> | PubMed | 86.96 | 87.25 | 87.55 | 87.05 |
>
> | GOUDA-BT | 0.2  | 0.4| 0.6 | 0.8 |
> |:--------| :---------:|:--------:|:--------:|:--------:|
> | Cora | 85.07 | 84.56 | 85.88 | 85.99 |
> | CiteSeer | 74.34 | 74.47 | 73.98 | 73.41 |
> | PubMed | 87.59 | 86.94 | 86.76 | 87.11 |
>
> ---
>
> ***Q4. As far as I know, the HSIC term is computationally intensive. Does this affect efficiency?***
>
> R4. There may be some misunderstandings. The HSIC term has a quadratic time complexity of $O(N^2)$ for the number of samples $N$. However, this complexity is mitigated to $O(K^2)$ in the proposed framework since HSIC is applied only to $K$ AC vectors instead of the entire node set. The proposed framework sets $K$ as a small number, which is independent of the size of the graph. Therefore, the employment of the HSIC term does NOT cause high computation in the proposed framework.
>
> ---
>
> ***Q5. Descriptions from Lines 143 to 153 are difficult to understand.***
>
> R5. We have carefully reviewed the content and have reorganized the passage for better clarity and comprehension. The passage now presents as follows.
>
> (1) Edge augmentation is to add or remove the augmented edges, which corresponds to inserting or dropping the nodes connected by these edges in the neighborhood of the impacted nodes. The shown edge removal will lead to dropping certain 2-hop neighbors (nodes 1, 4, 7, and 8) of node 0 during the aggregation phase.
>
> (2) Attribute augmentation is to replace the attributes of the impacted nodes with the altered attributes, which affects all nodes connected to them. The shown attribute augmentation can be seen as modifying the attributes of the augmented neighbors (nodes 1, 3, 4, 5, 6, 7, and 8) of node 0 during the aggregation phase.
>
> (3) Subgraph augmentation is to modify the specific subsets of the input graph (including its edges and attributes), which also can be seen as the perturbation in the neighborhoods of impacted nodes. The shown subgraph augmentation will lead to the removal of nodes 2, 4, 6, and 8 from the 2-hop neighborhood of node 0 during the aggregation phase.  Furthermore, node augmentation represents a specific case of subgraph augmentations, with the subset limited to a single node, thus rendering the aforementioned conclusion applicable to it.
>
> ---
>
> ***Q6. Reference [33] is based on adversarial training rather than adversarial attack.***
>
> R6. Thank you for the meticulous correction. We will correct the methodology of Reference [33] to adversarial training.
>
> ---
>
> ***Q7. Some commas are missing after equations, for instance, in Eq. (9).***
>
> R7. Thank you for your attention to detail. We will carefully check and comprehensively polish the manuscript to make sure that all formulas are followed by the correct punctuation.

---

> > ### Comment · Reviewer_9GrB · 2024-08-11
> >
> > Thank you for the response, which solves most of my concerns, and I will maintain my score on this paper.

---

### Official Review · Reviewer_BtAD · 2024-07-05

**Soundness:** 4
**Presentation:** 4
**Contribution:** 4
**Rating:** 7
**Confidence:** 4

**Summary:**

The paper reconsiders the formulation of graph augmentations in graph contrastive learning, introducing a novel perspective on GA through message passing. The proposed UGA framework interprets graph augmentations as mechanisms for aggregation and propagation between nodes, highlighting the significance of local aggregation and propagation within GA. The effectiveness of the proposed GOUDA framework is demonstrated in experimental evaluations.

**Strengths:**

- The formulation of GA from the perspective of message passing is intriguing. Construing GA as aggregation and propagation among neighbors adds significant flexibility.
- The proposed GOUDA framework demonstrates effectiveness and robustness.
- The paper exhibits superiority across various graph-based tasks, including node classification, node clustering, and graph classification. These results indicate that GOUDA is not limited to specific tasks.
- The computational complexity is significantly reduced over a wide range.
- Well-written.

**Weaknesses:**

- The paper lacks presentation on the interpretability of learned graph augmentation vectors. Given that graph augmentation can occur at nodes, edges, attributes, and sub-graphs, an explanation of how these learned vectors are interpreted is necessary.
- The paper should include a comparison of time consumption during experiments.

**Questions:**

- Can the proposed GOUDA, especially the Graph Augmentation Vectors (GAVs), be integrated with other Graph Contrastive Learning (GCL) methods?
- How to assess the contribution of the proposed GOUDA framework to Graph Contrastive Learning (GCL)?

**Limitations:**

The paper requires interpretability and should clarify the contribution to GCL.

---

> ### Author Rebuttal · Authors · 2024-08-07
>
> ***Q1. The paper lacks presentation on the interpretability of learned graph augmentation vectors.***
>
> R1. To provide interpretation for Graph Augmentation (GA), we would first introduce the interpretable explanations for GNNs (i.e., graph encoders) on graphs [1]. Within the encoder-relevant computation graph, i.e., the k-hop subgraph, a subgraph that is informative and most influential on the label is specified as an explanation. For a given node, the computation graph corresponds to its $k$-hop neighborhood; for a given graph, it represents the entire graph. Based on this introduction, GAs can be interpreted as techniques that seek to preserve the information-rich subgraphs. Our GA method, which interpolates a batch of Augmentation-Centered (AC) vectors into the input graph to emulate the GAs' effect, is proposed based on an observation: GA is equivalent to modifying the node attributes within the computation graphs of nodes, that is, local attribute perturbation. During the weight optimization guided by an error metric, these AC vectors adaptively capture task-related perturbation information from the computation graphs of nodes. Therefore, the learned AC vectors can be interpreted as the representations of subgraphs in the computation graph of nodes.
>
> [1] GNNExplainer: Generating Explanations for Graph Neural Networks. NeurIPS 2019
>
>
> ---
>
> ***Q2. The paper should include a comparison of time consumption during experiments.***
>
> R2. We understand your concerns about the complexity of the proposed framework. We have conducted a comprehensive comparison of the running time consumption of GCLs, as shown in Figure 3 of our manuscript. The accuracy and the running time for each training epoch are provided in the following table for your review.
>
> | Accuracy(%) / Time(s) | IMDB-BINARY  | IMDB-MULTI |
> |:--------| :---------:|:--------:|
> | Infograph | 73.03 / 0.82 | 49.69 / 0.89 |
> | GraphCL | 71.14 / 0.49 | 48.58 / 0.56 |
> | JOAO | 71.60 / 1.88 | 49.20 / 1.79 |
> | AD-GCL | 71.49 / 1.31 | 50.36 / 1.44 |
> | MVGRL | 74.20 / 1.16 | 51.20 / 1.02 |
> | GOUDA-IF (Ours) | 75.22 / 0.41 | 52.43 / 0.47 |
> | GOUDA-BT (Ours) | 76.80 / 0.46 | 53.05 / 0.55 |
>
> It is evident from the table that GOUDA not only outperforms the baselines but also consumes less time per epoch, demonstrating that GOUDA is an effective yet lightweight framework. The detailed analysis has been presented in Section 4.1 of our submitted manuscript.
>
> ---
>
> ***Q3. Can the proposed GOUDA, especially the Graph Augmentation Vectors (GAVs), be integrated with other Graph Contrastive Learning (GCL) methods?***
>
> R3. Of course. As a general framework, GOUDA possesses high compatibility with most GCLs, enabling integration through slight modifications of the graph encoder or the loss function. In our submitted manuscript, GOUDA is implemented as two models: GOUDA-IF, which utilizes InfoNCE loss, and GOUDA-BT, which employs BarlowTwins loss. Extensive experiments (such as graph classification in the table of R2) on various datasets and tasks demonstrate the outstanding performances of GOUDA-IF and GOUDA-BT, highlighting the broad compatibility and effectiveness of the framework GOUDA.
>
> ---
>
> ***Q4. How to assess the contribution of the proposed GOUDA framework to Graph Contrastive Learning (GCL)?***
>
> R4. We would like to illustrate the contribution of the proposed framework GOUDA to GCL from the following aspects:
> 1. **A novel perspective of graph augmentations**. GOUDA is designed with a thorough analysis of the mechanisms of existing Graph Augmentations (GAs) in GCLs. We provide a unified interpretation of these mechanisms from a message-passing perspective, offering new insights into the GCL field.
> 2. **A general graph augmentation module**. GOUDA presents a lightweight yet effective GA module, i.e., UGA, that achieves a theoretical unification of diverse GAs, providing an innovative GA strategy to GCLs.
> 3. **New SOTA**. The implemented models, GOUDA-IF and GOUDA-BT, have achieved a new state-of-the-art performance on many tasks, including node/graph classification.

---

> > ### Comment · Reviewer_BtAD · 2024-08-13
> >
> > Thank you for your response. My concerns have been addressed, so I have decided to maintain my original score.

---

### Official Review · Reviewer_i3ih · 2024-07-12

**Soundness:** 2
**Presentation:** 2
**Contribution:** 2
**Rating:** 5
**Confidence:** 3

**Summary:**

The paper introduces GOUDA, a versatile framework for Graph Contrastive Learning that addresses the limitations of existing graph augmentation techniques. GOUDA proposes a unified graph augmentation module capable of simulating various explicit graph augmentations, enhancing the generality and efficiency of GCL models. By incorporating both widely-adopted contrastive losses and a novel independence loss, GOUDA ensures consistency and diversity across different augmentations.

**Strengths:**

1. The paper addresses a crucial problem in graph contrastive learning: the specificity, complexity, and incompleteness of current graph augmentation techniques.
2. The paper conducts experiments on multiple datasets and compares the results with various baseline models.

**Weaknesses:**

1. Some of the mathematical derivations and formulas are complex and may be difficult for readers to follow. Although the notations are defined, the paper could provide more intuitive explanations and step-by-step derivations to enhance understanding.
2. While authors have compared with many GCL methods, the comparison with GA techniques like simple node augmentation is missing.

**Questions:**

1. How to interpolate the learned AC vectors?
2. How is k selected in line 181?
3. Graph augmentation is based on the assumption that the augmentation does not change the original class label, how does UGA guarantee this?

---

> ### Author Rebuttal · Authors · 2024-08-07
>
> ***Q1. The paper could provide more intuitive explanations and step-by-step derivations to enhance understanding.***
>
> R1. Thanks for your valuable suggestion. We will further clarify the mathematical formulas with intuitive explanations in the revised manuscript. Besides, please refer to the Appendix for detailed derivations.
>
> ---
>
> ***Q2. While authors have compared with many GCL methods, the comparison with GA techniques like simple node augmentation is missing.***
>
> R2. Simple node augmentation has been compared in our experiments since it is the augmentation strategy employed by many existing GCL methods, such as GraphCL [1]. Specifically, GraphCL employs random node dropping, a prevalent and simple node augmentation strategy, in the augmented graph construction. Following your suggestion, we will clarify this in the revised version.
>
> Figure 3 shows these comparisons from both effectiveness and efficiency perspectives. For convenience of review, the total accuracy and the running time for each training epoch are given as follows.
>
> | Accuracy(%) / Time(s) | IMDB-BINARY  | IMDB-MULTI |
> |:--------| :---------:|:--------:|
> | GraphCL | 71.14 / 0.49 | 48.58 / 0.56 |
> | GOUDA-IF (Ours) | 75.22 / 0.41 | 52.43 / 0.47 |
> | GOUDA-BT (Ours) | 76.80 / 0.46 | 53.05 / 0.55 |
>
> It is evident that GOUDA not only achieves superior performance but also possesses efficiency similar to GraphCL. This highlights that GOUDA is a lightweight yet highly effective framework.
>
> [1] Graph Contrastive Learning with Augmentations. NeurIPS 2020
>
> ---
>
> ***Q3. How to interpolate the learned AC vectors?***
>
> R3. The AC vectors are interpolated via a graph-based strategy. Besides the nodes in the original graph, AC vectors are treated as another type of nodes in the new graph. The additional connections (i.e., edges) from AC nodes to the original nodes are constructed according to the feature similarity, as formulated in Eq. 6. This interpolation strategy benefits both the reduction of the tuning parameters for graph augmentations and the flexibility by dynamically adjusting the varying importance of relationships.
>
> ---
>
> ***Q4. How is k selected in line 181?***
>
> R4. We recognize your concerns about the selection of the $k$, which is a key hyperparameter in the proposed GOUDA. The proposed framework sets $k$ as a small number, which is independent of the size of the graph. The submitted manuscript has presented an experimental assessment of the impact of varying $k$ on model performance, as shown in Figure 7, where $k$ is selected from the set {1, 5, 10, 20}. Our observations indicate that GOUDA is relatively insensitive to the hyperparameter $k$ as explained in Section 4.2. Therefore, one need not be overly concerned with the precise value of $k$ for GOUDA’s robustness to changes in $k$.
>
> ---
>
> ***Q5. Graph augmentation is based on the assumption that the augmentation does not change the original class label, how does UGA guarantee this?***
>
> R5. Note that NO data augmentation can guarantee semantic invariance in self-supervised learning since label information is unavailable. All the augmentation-based graph contrastive learning methods are based on the assumption of local smoothness in the space of graphs, and thus, slight changes between original and augmented graphs do not change the semantics, i.e., the label, of the graphs. Thus, the proposed UGA module, which unifies the four types of graph augmentations, has yet to guarantee the semantic invariance of augmentation. Fortunately, the learnable characteristic of UGA prevents the semantic shift from random operations in previous GAs. Thus, the invariance can be enhanced via the consistency constraints.

---

> > ### Comment · Reviewer_i3ih · 2024-08-09
> >
> > Thanks for authors' rebuttal, I have adjusted my score based on the response.

---

> > > ### Author Response · Authors · 2024-08-09
> > >
> > > Thanks for your feedback. We hope that our response appropriately answers your questions. We are willing to provide further clarification if you have any additional concerns.

---

### Comment · Area_Chair_zP8L · 2024-08-09
**Discussion period instructions**

Dear Reviewers,

The authors have provided comprehensive rebuttals and tried to address the concerns raised in your reviews. Please take the time to review their responses carefully. If you have any further questions or require additional clarification, please engage in a discussion with the authors. Thank you for your continued efforts.

AC

---

### Decision · Program_Chairs · 2024-09-25

**Decision:**

Accept (poster)

**Comment:**

This paper addresses potential limitations in existing Graph Contrastive Learning (GCL) frameworks, specifically the issues of specificity, complexity, and incompleteness of Graph Augmentation (GA) techniques. The authors introduce UGA, a method with theoretical guarantees, alongside GOUDA, a framework that can be integrated with various contrastive losses. Experimental results demonstrate the effectiveness of GOUDA in enhancing GCL performance.

The paper has received predominantly positive feedback both before and after the rebuttal period, with final scores of (8, 7, 7, 5). Please carefully consider the questions and concerns raised by the reviewers, particularly those from reviewer i3ih, when the authors preparing the final version of the paper, e.g., some implementation details and more comparisons.